# 🍓 Mulberry: Empowering MLLM with o1-like Reasoning and Reflection via Collective Monte Carlo Tree Search

**Huanjin Yao**[2,3,*], **Jiaxing Huang**[1,*,✉], **Wenhao Wu**[3], **Jingyi Zhang**[1], **Yibo Wang**[2], **Shunyu Liu**[1], **Yingjie Wang**[1], **Yuxin Song**[3], **Haocheng Feng**[3], **Li Shen**[4], **Dacheng Tao**[1,✉]

[1] Nanyang Technological University [2] Tsinghua University [3] Baidu Inc. [4] Sun Yat-sen University

[*] Equal Contribution    [✉] Corresponding Author

## Abstract

In this work, we aim to develop an MLLM that understands and solves questions by learning to create each intermediate step of the reasoning involved till the final answer. To this end, we propose Collective Monte Carlo Tree Search (CoMCTS), a new learning-to-reason method for MLLMs, which introduces the concept of collective learning into "tree search" for effective and efficient reasoning-path searching and learning. The core idea of CoMCTS is to leverage collective knowledge from multiple models to collaboratively conjecture, search and identify effective reasoning paths toward correct answers via four iterative operations including Expansion, Simulation and Error Positioning, Backpropagation, and Selection. Using CoMCTS, we construct Mulberry-260k, a multimodal dataset with a tree of rich, explicit and well-defined reasoning nodes for each question. With Mulberry-260k, we perform collective SFT to train our model, Mulberry, a series of MLLMs with o1-like step-by-step Reasoning and Reflection capabilities. Extensive experiments demonstrate the superiority of our proposed methods on various benchmarks. Code is available at `https://github.com/HJYao00/Mulberry`.

## 1 Introduction

*"What I cannot create, I do not understand."*

*—Richard Feynman*

Multimodal large language models (MLLMs) embody the essence of this dictum, which understand the world by learning to create expected responses to multimodal inputs such as images and text. While MLLMs have recently shown significant progress in straightforward tasks [1, 2], they often experience obviously increased failures on complex tasks requiring in-depth reasoning [3]. Feynman's dictum might be the perfect metaphor of such failures of MLLMs, as we should only be able to work something out if we can create and have a firm understanding of each step of the reasoning involved. However, current MLLMs predominantly operate in a simple "direct prediction" mode [4], *i.e.*, generating brief, final answers to questions with little explicit and well-defined intermediate reasoning steps.

In this work, we aim to develop an MLLM that understands and solves questions by learning to create each intermediate step of the reasoning involved till the final answer. Recent advances in NLP, such as OpenAI o1 [5], have shown great potential in enabling LLM to learn to reason and tackle complex language tasks [6]. The core design of these advances lies in AlphaGo-like "tree search": they employ tree search methods, like MCTS [7], to bootstrap an LLM itself to build a tree

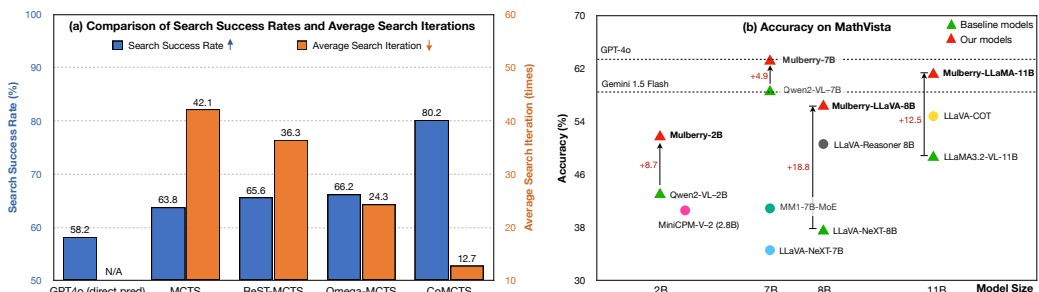

Figure 1: **(a)** Our CoMCTS shows great superiority in search effectiveness and efficiency against other tree search methods. **(b)** Our Mulberry, trained on CoMCTS-searched data, outperforms most open-sourced MLLMs and achieves competitive results against closed-source ones, showing outstanding abilities in step-by-step reasoning and reflection.

of intermediate thoughts, explore effective reasoning paths, and leverage these paths to teach model to reason step-by-step.

An intuitive idea is to directly apply these "tree search" methods to search effective reasoning paths for MLLMs, which, however, does not work well. As shown in Figure 1, we believe this is largely attributed to several search challenges for MLLMs. (1) *Search Effectiveness:* Traditional MCTS methods [7, 8, 9, 10] generally work by self-bootstrapping while current MLLMs are typically trained with little explicit and well-defined intermediate reasoning steps, making these search methods often trapped in homogeneous low-quality nodes within the reasoning space of a single MLLM, ultimately leading to low search success rates. (2) *Search Efficiency:* Traditional MCTS methods typically expand and explore only one reasoning node per search iteration, which advance a single step each time and demand massive iterations, making them inefficient for computation-intensive MLLMs.

To tackle these challenges, we propose Collective Monte Carlo Tree Search (CoMCTS), a new learning-to-reason method for MLLMs, which introduces the concept of collective learning into "tree search" for effective and efficient reasoning-path searching and learning. The core idea of CoMCTS is to leverage collective knowledge to collaboratively conjecture, search and identify effective reasoning paths toward correct answers. Specifically, CoMCTS searches effective reasoning paths iteratively, and in each iteration, it leverages collective knowledge from multiple MLLMs to jointly (a) expand diverse and complementary candidate subsequent reasoning nodes till the end from a given start node, (b) simulate reasoning outcomes, position error candidate nodes and prune them along with their child nodes, (c) backpropagate to update the score and visit count of each reasoning node in a bottom-up manner, and (d) select the leaf reasoning node with the highest Upper Confidence Bound value as next start node.

In this way, our CoMCTS achieves effective and efficient reasoning search. (1) The joint expansion mechanism enables CoMCTS to concatenate reasoning trajectories from multiple MLLMs via iterative search, ultimately constructing an unified reasoning tree comprising diverse and complementary reasoning nodes. Thus, it allows reasoning-path search not only within the reasoning space of a given MLLM itself but also among those of others, benefiting from the synergy of multiple MLLMs while avoiding being trapped in homogeneous low-quality nodes within the reasoning space of a single MLLM itself. (2) The joint simulation and error positioning mechanism enables CoMCTS to, in each search iteration, skip multiple intermediate steps and select the last correct step as the next start node, largely reducing search time while maintaining search effectiveness. Here, collective knowledge is also crucial as it is often challenging for a model to recognize and position errors made by itself while relatively easy by using other models.

Furthermore, we extend our CoMCTS for reflective reasoning-path search. Based on the unified reasoning tree constructed by CoMCTS, which provides both positive and negative reasoning nodes , we identify and integrate negative sibling nodes into effective reasoning paths to build the reflective reasoning path that includes a transition from a negative reasoning node to a positive one. By learning from reflective reasoning paths, MLLMs can perform appropriate step-wise reflection, dynamically calibrating their reasoning trajectory from an erroneous node toward a correct one during long-chain reasoning. Here, collective knowledge facilitates reflective reasoning-path search by providing a rich set of diverse positive and negative reasoning nodes.

Using our CoMCTS, we search effective and reflective reasoning paths for a set of multimodal inputs, and construct Mulberry-260k, a Multimodal learning-to-Reason-and-Reflect dataset with a tree of rich, explicit and well-defined reasoning nodes for each question. With Mulberry-260k, we perform collective supervised fine-tuning to train our model, Mulberry, a series of Multimodal LLMs with o1-like step-by-step Reasoning and Reflection capabilities.

The main contributions of this work are fourfold. ***First***, we introduce the concept of collective learning into MCTS, and propose CoMCTS which leverages collective knowledge to collaboratively conjecture, search and identify effective and reflective reasoning paths for MLLMs, significantly improving search effectiveness and efficiency. To the best of our knowledge, this is the first work that explores collective learning with MCTS for MLLMs. ***Second***, we construct Mulberry-260k that provides a valuable resource for advancing research in step-by-step reasoning and reflection in MLLMs. ***Third***, we develop Mulberry, a series of MLLMs with outstanding capabilities in step-by-step reasoning and reflection. ***Fourth***, extensive experiments demonstrate the superiority of our proposed methods on various benchmarks.

## 2 Related Work

### 2.1 Multimodal Large Language Model

MLLMs [1, 2, 11, 12, 13, 14, 15, 16] have made notable advancements in general vision-language understanding, enabling them to interpret visual semantics across various domains. Recent studies [17, 3] explore MLLM reasoning and reveal that directly employing CoT prompt to derive the final answer may result in limited gains or even degradation. In addition, some studies [18, 19] introduce plan-based CoT prompting to guide models to generate intermediate information for predicting final answers. Recent advances [4] attempt structured reasoning with a planed flow of certain pre-defined stages, enhancing the CoT capabilities [15] of MLLMs. Differently, this paper, for the first time, introduces the concept of "tree search" into MLLM reasoning and proposes a novel CoMCTS technique to search effective and reflective reasoning paths to train our Mulberry, a series of MLLMs with outstanding capabilities in step-by-step reasoning and reflection.

### 2.2 Large Language Model Reasoning

LLM reasoning methods can be broadly categorized into three types, *i.e.*, prompt-based, plan-based and learning-based reasoning. Prompt-based methods, like Chain-of-Thought (CoT) [20], mimic human reasoning by providing a few hand-crafted, step-by-step solutions as references. Plan-based methods, such as Tree/Graph-of-thought [21, 22], predict multiple reasoning paths in a tree or graph manner and take consistent units of thought for thoughtful decision-making. Learning-based reasoning methods, represented by GPTo1, Star [23], Iter-MCTS [6] and ReST-MCTS [24], first employ tree search approaches [25], like MCTS, to bootstrap an LLM itself to build a tree of intermediate thoughts, explore effective reasoning paths, and leverage these paths to train model to reason step-by-step.

### 2.3 Monte-Carlo Tree Search

Monte-Carlo Tree Search (MCTS) is a powerful search paradigm for complex decision making problems and has been extensively explored across diverse fields, including games [26, 27], robotics [28, 29], theorem proving [30], matrices multiplication [31], etc. For instance, AlphaGo [26] introduces deep learning into MCTS, achieving superhuman results in board and video games [26, 27]. Besides, [32, 33] explore MCTS for path finding and train timetabling problems, while [34] integrates MCTS into physics-informed planning networks for robot control. In this work, we propose CoMCTS that enables effective and reflective reasoning-path searching and learning on MLLMs.

### 2.4 Collective Learning

Collective learning, also known as Co-training, aims to harness collective intelligence of multiple individuals to improve learning outcomes. This concept originates in early pioneering studies [35, 36, 37], which utilize collective knowledge to address data insufficiency issues in classification learning. Recent advances introduce collective learning into deep neural networks for efficient and effective

deep learning. For example, [38, 39] employ collective knowledge from multiple classifiers to predict more accurate pseudo-labels for semi-supervised classification; [40] utilizes collective knowledge from multiple discriminators to enhance image discrimination and generation; and [41] leverages the synergy of multiple models for reinforcement learning.

# 3    Method

We first present CoMCTS that introduces collective learning into "tree search" for effective and efficient reasoning-path searching and learning. We then illustrate the extension of CoMCTS for reflective reasoning-path search, and describe data construction and model training using CoMCTS.

## 3.1    CoMCTS for effective reasoning

The core idea of CoMCTS is to leverage collective knowledge to collaboratively conjecture, search and identify effective reasoning nodes in an iterative manner, aiming to find effective reasoning paths leading to correct answers.

We denote a policy model as $\pi$, initialized by a pre-trained MLLM. We leverage collective knowledge from a group of MLLMs $\{\pi_1, \pi_2, ..., \pi_K\}$ to jointly search and learn effective reasoning paths. Given a multimodal input question $Q$ (*e.g.*, a text instruction with an image, $Q = \{\text{text}, \text{image}\}$), each model $\pi$ can generate a sequence of intermediate reasoning states toward the final answer $(s_1, s_2, s_3, ..., s_M) \sim \pi_\theta(\cdot|Q)$ via autoregressive next token prediction. We define the intermediate reasoning state at step $m$ as $s_m$ and the state generated by model $\pi_k$ at step $m$ as $s_m^k$. Each reasoning step consists of one or a few sentences.

CoMCTS algorithm begins at the root node, *i.e.*, either the start of a response or an incomplete response, and performs reasoning-path search via a certain number of iterations, where each iteration comprises four key operations: (a) Expansion, (b) Simulation and Error Positioning, (c) Backpropagation, and (d) Selection, as elaborated below.

**(a) Expansion.** The goal of this operation in CoMCTS is to expand the current leaf reasoning node (if it is not a terminal node) to integrate new subsequent candidate reasoning nodes. Given the current leaf node $s_m^k$ (*i.e.*, the node selected by Operation (d) Selection or the root node), CoMCTS utilizes collective knowledge from a group of MLLMs, $\{\pi_1, \pi_2, ..., \pi_K\}$, to jointly expand a set of diverse and complementary candidate reasoning paths $S_{\text{candidate}} = \cup_{j=1}^K S_{\text{candidate}}^j$ in parallel till terminal node:

$$S_{\text{candidate}}^j \sim \pi_j(\cdot|Q, \text{Parent}(s_m^k), s_m^k), \tag{1}$$

where $\text{Parent}(s_m^k)$ returns all parent nodes of $s_m^k$ and $(\text{Parent}(s_m^k), s_m^k)$ denotes the current reasoning path from the root node to $s_m^k$. $S_{\text{candidate}}^j = \{s_i^j\}$ stands for a potential reasoning path generated by model $\pi_j$ starting from $s_m^k$.

**(b) Simulation and Error Positioning.** In this operation, CoMCTS utilizes collective knowledge from $\{\pi_1, \pi_2, ..., \pi_K\}$ to jointly estimate the potential value of child nodes $s_i^j \in S_{\text{candidate}}$ (added in Operation (a)), and considers low-score nodes as erroneous reasoning nodes, and positions and filters out them along with their child nodes:

$$R(s_i^j) = \frac{1}{K} \sum_{l=1}^K \pi_l(\cdot|\text{prompt}_{\text{eval}}, Q, \text{Parent}(s_i^j), s_i^j) \tag{2}$$

$$S_{\text{candidate}}^* = \{s_i^j \in S_{\text{candidate}} | R(s_i^j) >= t\} \tag{3}$$

where $R(s_i^j)$ denotes a reasoning node evaluation function that uses the prompt, $\text{prompt}_{\text{eval}}$, to request a group of MLLMs, $\{\pi_1, \pi_2, ..., \pi_K\}$, to jointly evaluate the candidate reasoning node $s_i^j$. $t$ is a threshold and discontinued reasoning nodes in $S_{\text{candidate}}^*$ are automatically removed following the error node removal in Eq.(3).

**(c) Backpropagation.** Given the new reasoning tree expanded and simulated using collective knowledge in Operations (a)-(b), CoMCTS performs a bottom-up update from the leaf nodes back to the root node. Each node $s$ along the newly expanded path in the reasoning tree updates its statistics,

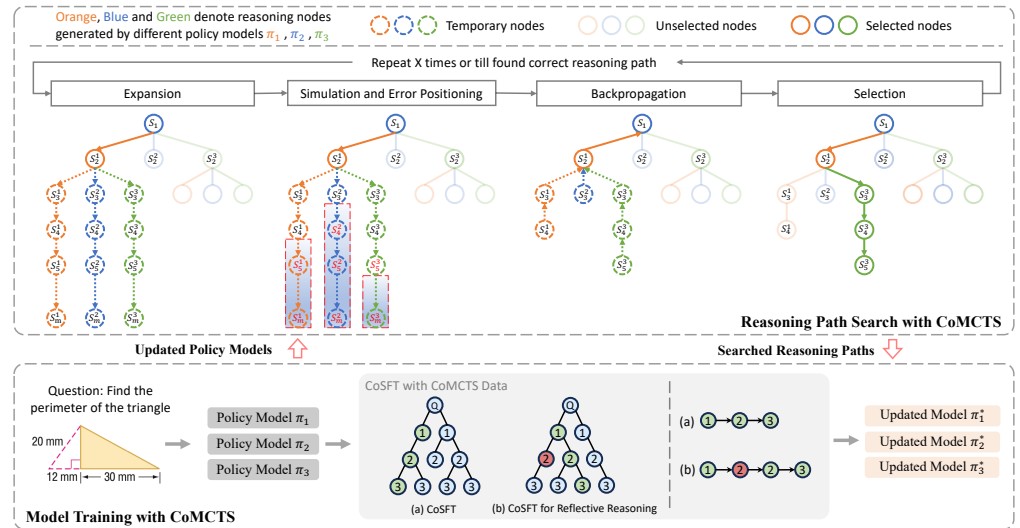

Figure 2: **Overview.** Our CoMCTS trains Mulberry with two alternating phases. *In top part,* CoMCTS searches reasoning paths iteratively, and in each iteration, it utilizes collective knowledge from multiple MLLMs to jointly (a) expand diverse and complementary candidate subsequent reasoning nodes till the end from a given start node, (b) simulate reasoning outcomes, position error candidate nodes and prune them along with their child nodes, (c) backpropagate to update the score and visit count of each reasoning node in a bottom-up manner, and (d) select the leaf reasoning node with the highest UCB value as next start node. *In bottom part,* we train the model to learn from the reasoning trees constructed by CoMCTS.

including visit count $N$ and node value $V$:

$$V(s) \leftarrow \frac{N(s) \cdot V(s) + \sum_{s_l \in \text{Child}(s)} R(s_l)}{N(s) + \text{CountChild}(S^*_{\text{candidate}}, s)}, \tag{4}$$

$$N(s) \leftarrow N(s) + \text{CountChild}(S^*_{\text{candidate}}, s), \tag{5}$$

where $\text{Child}(s)$ returns all the child nodes of $s$, and $\text{CountChild}(S^*_{\text{candidate}}, s)$ is a child node counting function that calculates the number of child nodes of $s$ in $S^*_{\text{candidate}}$.

**(d) Selection.** Following Operations (a), (b) and (c), CoMCTS traverses the updated reasoning tree to select the next starting node. This selection is guided by the Upper Confidence Bound (UCB) value, which balances search exploration and exploitation. The UCB value of a node $s$ is computed using the node reward value $V(s)$ and the visit count $N(s)$. Among the candidate nodes $s \in S^*_{\text{candidate}}$, the one with the highest UCB value is chosen as the starting node $s_m^{k^*}$ for next search iteration:

$$s_m^{k^*} = \underset{s \in S^*_{\text{candidate}}}{\arg\max} V(s) + c \cdot \sqrt{\frac{\log N(\hat{s})}{1 + N(s)}} \tag{6}$$

where $c$ stands for a constant which controls the level of exploration. $\hat{s}$ denotes the parent node of $s$.

**CoMCTS.** These four operations, *i.e.*, (a) Expansion, (b) Simulation and Error Positioning, (c) Backpropagation and (d) Selection, are repeated for a pre-defined number of iterations or until correct reasoning paths are found. This iterative process allows CoMCTS to construct a question-dependent reasoning tree $S$ with the correct reasoning path $Y$, and ultimately form a multimodal learning-to-reason data triplet $\{Q, Y, S\}$. By applying our CoMCTS to a set of multimodal questions, we can construct a collection of multimodal learning-to-reason data triplets, which provide a tree of rich, explicit and well-defined reasoning nodes toward the final answer for each question and enable MLLMs to learn to reason step-by-step.

### 3.2 CoMCTS for reflective reasoning

In this subsection, we extend CoMCTS for reflective reasoning-path search. Based on the unified reasoning tree constructed by CoMCTS, *i.e.*, $\{Q, Y, S\}$, which provides both positive and negative

reasoning nodes, we identify and integrate negative sibling nodes into effective reasoning paths to build the reflective reasoning path that includes a transition from a negative reasoning node to a positive one.

**Identifying negative sibling node.** Given the effective reasoning path $Y$, we identify the negative sibling reasoning node for $s \in Y$ using UCB:

$$s_{\text{neg}} = \underset{s_l \in \text{Sibling}(s)}{\arg\min} \; \text{UCB}(s_l) - \text{UCB}(s), \;\; \forall s \in Y, \tag{7}$$

where $\text{Sibling}(s)$ returns all the sibling nodes of $s$, *i.e.*, the nodes on the same hierarchical level under the same parent node of $s$. $\text{UCB}(s) = V(s) + c \cdot \sqrt{\frac{\log N(\hat{s})}{1+N(s)}}$ as in Eq. 6.

**Constructing reflective reasoning path.** Based on Eq. 7, we randomly sample a reasoning node $s \in Y$ with its negative sibling node $s_{\text{neg}}$, and concatenate them with a reflection prompt to form a reflection trajectory, *i.e.*, $(s_{\text{neg}}, \text{prompt}_{\text{reflect}}, s)$. We then use a function $\text{Replace}(\cdot)$ that replaces $s \in Y$ with $(s_{\text{neg}}, \text{prompt}_{\text{reflect}}, s)$ to convert $Y$ into the reflective reasoning path $Y_{\text{reflect}}$:

$$Y_{\text{reflect}} = \text{Replace}(Y, s, (s_{\text{neg}}, \text{prompt}_{\text{reflect}}, s)), \tag{8}$$

where $\text{prompt}_{\text{reflect}}$ denotes a reflection prompt, such as "The previous reasoning step is wrong and let's rethink it again." Then, we can integrate the reflective reasoning path $Y_{\text{reflect}}$ into our data as a quadruplet $\{Q, Y, Y_{\text{reflect}}, S\} \in D$.

**Collective Supervised Fine-Tuning (CoSFT).** Given $(Q, Y) \in \mathcal{D}$, we apply standard SFT objective to train our MLLM to learn from $D$ constructed by CoMCTS:

$$\mathcal{L}_{\text{CoSFT}}(\pi_k) = \sum_{(Q,Y) \in \mathcal{D}} \log \pi_k(Y|Q), \tag{9}$$

where $Y = \{s\}$ denotes the effective reasoning path that includes a sequence of reasoning nodes collectively conjectured, searched and identified by a group of MLLMs.

**CoSFT for reflective reasoning**. Given a question and its reasoning tree $(Q, S) \in \mathcal{D}$ constructed by CoMCTS, we randomly sample a reflective reasoning path $Y_{\text{reflect}}$ from $S$ as in Eqs.7-8, and conduct CoSFT for reflective reasoning:

$$\mathcal{L}_{\text{CoSFT-Re}}(\pi_k) = \sum_{(Q,Y_{\text{reflect}}) \in \mathcal{D}} \log \pi_k(Y_{\text{reflect}}|Q), \tag{10}$$

where $Y_{\text{reflect}} = \{s\}$ denotes the reflective reasoning path that includes an additional step-wise reflection trajectory.

The goal of $\mathcal{L}_{\text{CoSFT}}$ and $\mathcal{L}_{\text{CoSFT-Re}}$ is to maximize the log probability of effective and reflective reasoning path $Y$ and $Y_{\text{reflect}}$ over a tree of reasoning nodes $S$ generated by CoMCTS. In addition, $\mathcal{L}_{\text{CoSFT-Re}}$ enables to leverage the negative information during CoMCTS search process by learning to calibrate negative reasoning nodes.

### 3.3 Training with Collective MCTS

Using CoMCTS, we search effective and reflective reasoning paths for a set of multimodal input questions, and construct Mulberry-260k, a multimodal learning-to-reason-and-reflect dataset with a tree of rich, explicit and well-defined reasoning nodes for each question, *i.e.*, a set of quadruplets $\{Q, Y, Y_{\text{reflect}}, S\} \in D$. To learn collective knowledge from Mulberry-260k, we perform collective SFT to train our model, Mulberry, a series of Multimodal LLMs with o1-like step-by-step Reasoning and Reflection capabilities.

---

**Algorithm 1** Training Mulberry with CoMCTS

**Input:** a set of policy models $\{\pi_1, \pi_2, ..., \pi_K\}$ initialized by different MLLMs; a set of multimodal questions $D_Q$

**for** *i = 1 to MaxEpoch* **do**

    Reasoning Tree Search using CoMCTS:

    **for** $Q \in D_Q$ **do**

        Collective Monte Carlo tree search:

        $\{Q, Y, S\} = \text{CoMCTS}(\{\pi_1, \pi_2, ..., \pi_K\}; Q)$

        **if** *found an effective reasoning path* **then**

            Search and find $Y_{\text{reflect}}$ from $S$

            Add $\{Q, Y, Y_{\text{reflect}}, S\}$ into $D$

            Remove $Q$ from $D_Q$

    Model Training with CoMCTS Reasoning Trees:

    **for** *k = 1 to K* **do**

        **for** $(Q, Y, Y_{\text{reflect}}, S) \in D$ **do**

            Supervised Fine-Tuning:

            Optimize $\pi_k$ via $\mathcal{L}_{\text{CoSFT}}(\pi_k)$ and $\mathcal{L}_{\text{CoSFT-Re}}(\pi_k)$

**Output:** Trained policy models $\{\pi_1, \pi_2, ..., \pi_K\}$

---

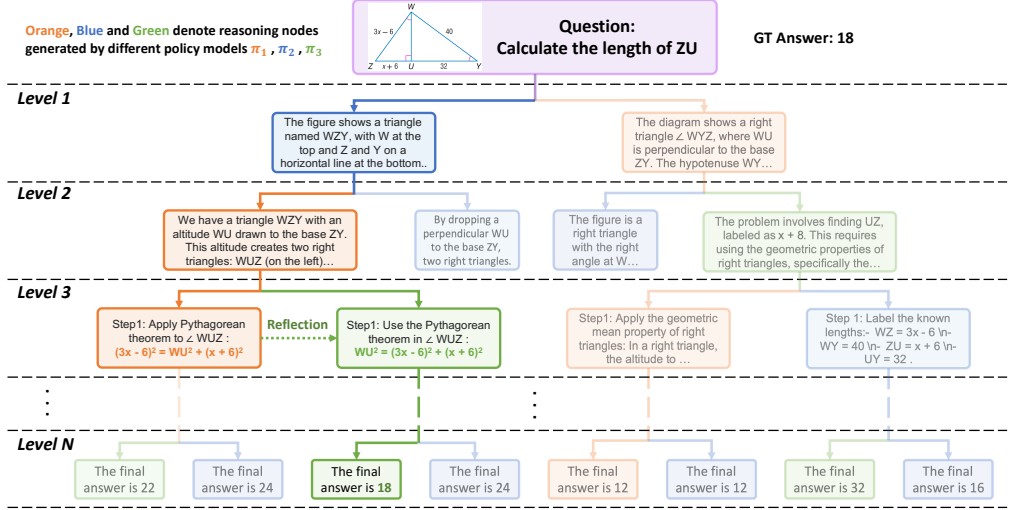

Figure 3: Qualitative illustration of reasoning tree searched by CoMCTS with rich, explicit, well-defined reasoning nodes.

## 4 Experiments

In this section, we first introduce our CoMCTS-generated dataset, Mulberry-260K, including its sources, construction, and analysis in Sec. 4.1, and provide implementation details in Sec. 4.2. We then present the main results in Sec. 4.3, demonstrating the effectiveness of the searched data (*i.e.*, Mulberry-260K) and the trained models (*i.e.*, Mulberry). In Sec. 4.4, we perform comprehensive ablation studies on the impact of effective and reflective reasoning data and the contributions of collective knowledge sources. Sec. 4.5 discusses the effectiveness and efficiency of CoMCTS with other tree search methods.

### 4.1 Dataset

**The Sources of Raw Data.** To construct a comprehensive and general-purpose tree-based reasoning dataset, we collect 260K raw multimodal input questions (*i.e.*, a text task instruction with an image as an input question) from a wide range of domains, covering General Multimodal Understanding, Mathematics, Figure Understanding, Realworld Understanding, Science, Medical Image Understanding, etc. The specific data sources are provided in the Appendix I.

**Reasoning Data Construction.** As detailed in Sec. 3 and Algorithm 1 and visually illustrated in Figures 2 and 3, we employ our CoMCTS to search effective and reflective reasoning paths for a set of raw multimodal input questions as collected from the mentioned "The Sources of Raw Data", ultimately constructing our dataset, Mulberry-260K. Note we only sample 15K data for reflective reasoning training to avoid overabundance of reflection data.

### 4.2 Implementation Detail

We implement collective learning in CoMCTS with four models, including GPT-4o, Qwen2-VL-7B, LLaMA-3.2-11B-Vision-Instruct, and Qwen2-VL-72B, to construct Mulberry-260K. In CoMCTS, we set maximum search iteration to 20 and threshold $t$ in Eq. 3 to 0. In each iteration, each model generates one candidate reasoning path to balance search exploration and exploitation. We adopt four popular MLLMs as baselines, conducting experiments on Qwen2-VL-7B and LLaMA-3.2-11B-Vision-Instruct to examine the effectiveness of CoMCTS, and on Qwen2-VL-2B and LLaVA-NeXT-8B to study the generalization of CoMCTS-searched data. Training details are in the Appendix B.

### 4.3 Main Results

To examine the effectiveness of searched data (*i.e.*, Mulberry-260K) and trained models (*i.e.*, Mulberry), we conduct extensive experiments with four powerful baseline models, and comprehensively benchmark our Mulberry with various state-of-the-arts, including general and reasoning-based

Table 1: **Main Results.** To examine the effectiveness of the searched data (*i.e.*, Mulberry-260K) and the trained models (*i.e.*, Mulberry), we conduct extensive experiments with four powerful baseline models, and comprehensively benchmark our Mulberry with various state-of-the-arts, including general and reasoning-based MLLMs.

| Method | MathVista | MMStar | MMMU | ChartQA | DynaMath | HallBench | MM-Math | $MME_{sum}$ | AVG |
|---|---|---|---|---|---|---|---|---|---|
| *Closed-Source Model* | | | | | | | | | |
| GPT-4o [42] | 63.8 | 63.9 | 69.1 | 85.7 | 63.7 | 55.0 | 31.8 | 2329 | 64.5 |
| Claude-3.5 Sonnet [43] | 67.7 | 62.2 | 68.3 | 90.8 | 64.8 | 55.0 | - | 1920 | - |
| *Open-Source Model* | | | | | | | | | |
| MM-1.5-7B [44] | 47.6 | - | 41.8 | 78.6 | - | - | - | 1861 | - |
| Idefics3-LLaMA3-8B [45] | 58.4 | 55.9 | 46.6 | 74.8 | - | - | - | 1937 | - |
| InternVL2-8B [46] | 58.3 | **61.5** | 51.8 | 83.3 | 39.7 | - | - | 2210 | - |
| MiniCPM-V-2.6-8B [47] | 60.6 | 57.5 | 49.8 | - | - | 48.1 | - | 2348 | - |
| DeepSeek-VL2-MOE-4.5B [48] | 62.8 | 61.3 | 51.1 | 86.0 | - | - | - | 2253 | - |
| *Reasoning Model* | | | | | | | | | |
| LLaVA-CoT-11B [4] | 54.8 | 57.6 | - | - | - | 47.8 | - | - | - |
| LLaVA-Reasoner-8B [3] | 50.6 | 54.0 | 40.0 | 83.0 | - | - | - | - | - |
| Insight-V-8B [49] | 49.8 | 57.4 | 42.0 | 77.4 | - | - | - | 2069 | - |
| LLaVA-NeXT-8B [50] | 37.5 | 42.1 | 41.7 | 69.5 | 22.7 | 33.4 | 0.6 | 1957 | 39.7 |
| Mulberry-LLaVA-8B | 56.3 | 54.5 | 43.0 | 79.5 | 34.1 | 47.5 | 18.9 | 2021 | 50.7[11↑] |
| Llama-3.2-11B-V-Ins. [51] | 48.6 | 49.8 | 41.7 | 83.4 | 34.3 | 40.3 | 4.1 | 1787 | 45.8 |
| Mulberry-Llama-11B | 61.1 | 58.5 | 45.6 | 83.5 | 37.2 | 48.9 | 18.7 | 2035 | 53.3[7.5↑] |
| Qwen2-VL-2B [2] | 43.0 | 48.0 | 41.1 | 73.5 | 24.9 | 41.7 | 1.0 | 1872 | 42.5 |
| Mulberry-2B | 51.7 | 51.3 | 42.0 | 77.7 | 30.0 | 44.9 | 13.9 | 2013 | 47.9[5.4↑] |
| Qwen2-VL-7B [2] | 58.2 | 60.7 | 54.1 | 83.0 | 42.1 | 50.6 | 5.9 | 2327 | 54.7 |
| Mulberry-7B | **63.1** | 61.3 | **55.0** | **83.9** | **45.1** | **54.1** | 23.7 | 2396 | **58.9**[4.2↑] |

MLLMs. The evaluation spans 8 widely used and challenging datasets, covering the fields ranging from general and mathematical reasoning to hallucination and multi-disciplinary understanding and reasoning in Tab. 1.

**Comparison with baselines.** We first conduct experiments on baselines Qwen2-VL-7B and LLaMA-3.2-11B-Vision-Instruct that are involved in collective learning of CoMCTS for joint reasoning-path conjecture, search and identification. Trained with jointly-searched Mulberry-260k data, Mulberry-7B and Mulberry-11B bring clear performance improvements against their baselines, *i.e.*, +4.2% over Qwen2-VL-7B and +7.5% over LLaMA-3.2-11B-Vision-Instruct averaged on 8 benchmarks, validating the effectiveness of CoMCTS. On the other hand, we examine the generalization of Mulberry-260k by applying it to train other models not involved in CoMCTS, such as Qwen2-VL-2B and LLaVA-NeXT-8B. Trained with Mulberry-260k, Mulberry-2B and Mulberry-8B enhance their baselines with +5.4% and +11.0% gains, respectively, averaged on 8 benchmarks, demonstrating the generalization of CoMCTS.

**Comparison with reasoning-response models.** We benchmark Mulberry with various state-of-the-art reasoning-response models. Using the same base model LLaVA-NeXT-8B [50], our Mulberry outperforms LLaVA-Reasoner-8B and Insight-V-8B by +5.7% and +6.5% on mathematical benchmark MathVista, and by +3.0% and +1.0% on multi-disciplinary benchmark MMMU, respectively. Besides, Mulberry-11B surpasses LLaVA-COT-11B by +6.3% on MathVista under the same baseline LLaMA-3.2-11B-Vision-Instruct. The great superiority of Mulberry is largely attributed to our CoMCTS, which conducts collective tree search and generates rich, explicit and well-defined reasoning nodes with flexible numbers of steps.

**Comparison with state-of-the-arts.** We compare our Mulberry with popular state-of-the-art models, both open-source and closed-source. The results in Tab. 1 show that Mulberry, trained on CoMCTS-searched data, outperforms most open-sourced MLLMs and achieves competitive results against closed-source ones, demonstrating outstanding abilities in step-by-step reasoning and reflection.

## 4.4 Ablation Study

**Ablation Study on CoMCTS.** We conduct ablation studies with powerful GPT-4o as the baseline over 1K samples from Geo3K [52] and GeoQA-Plus [53], as shown in Tab. 2. As the core of CoMCTS, we examine how each model in the collective learning group contributes to the overall tree search performance. Tab. 2 reports the Search Success Rates, and baseline GPT-4o works not

Table 2: **Ablation Study on CoMCTS.** We study how each model in CoMCTS collective learning contribute to overall tree search performance in Search Success Rate (S.S.R.).

| Direct Pred | CoMCTS | | | | S.S.R. |
|---|---|---|---|---|---|
| GPT-4o | GPT-4o | Qwen2-VL-7B | Llama3.2-11B | Qwen2-VL-72B | |
| ✔ | | | | | 58.2 |
| | ✔ | | | | 63.8 |
| | ✔ | ✔ | | | 66.2 |
| | ✔ | ✔ | ✔ | | 69.7 |
| | ✔ | ✔ | ✔ | ✔ | 80.2 |

Table 3: **Ablation Study on Mulberry.** As Mulberry is trained with effective and reflective reasoning data searched by CoMCTS, we study their respective contributions.

| Methods | MathVista |
|---|---|
| Baseline | 43.0 |
| w/o Reflection Data (+245K) | 50.9 |
| w/ Reflection Data (+15K) | 51.7 |

very well without tree search. It shows that CoMCTS with only GPT-4o improves the performance to 63.8%, largely because our tree search designs like expansion, simulation and error positioning can work even without using collective knowledge. Besides, progressively involving more models into CoMCTS consistently improves the search performance, even with small models like Qwen2-VL-7B (*i.e.*, +2.4%), demonstrating the effectiveness of CoMCTS in capturing useful collective knowledge not only with large models but also from small models. In final, the inclusion of all four models in the proposed CoMCTS performs clearly the best, *i.e.*, 80.2%, validating the effectiveness of collective learning on reasoning tree search.

**Ablation Study on Mulberry.** We train Mulberry with effective and reflective reasoning data searched by CoMCTS and study their respective contributions to overall reasoning performance. Results in Tab. 3 on MathVista show that incorporating reflection data enhances performance by 0.8%, demonstrating the complementarity of effective and reflective reasoning data searched by CoMCTS.

## 4.5 Discussion

**Comparison with other tree search methods.** We compare our CoMCTS with other tree search methods in search effectiveness and efficiency, including the baseline "GPT-4o direction prediction", "traditional MCTS [7]", "ReST-MCTS [24]" that enhances MCTS by introducing partial search, and "Omega-MCTS [9]" that improves MCTS by designing binary search. Tab. 4 shows the results in search success

Table 4: **Comparison with other tree search methods.** "GPT-4o (direct)" refers to baseline without tree search. CoMCTS outperforms in search effectiveness and efficiency.

| Methods | Search Success Rate ↑ | Average Search Iteration ↓ |
|---|---|---|
| GPT4o (direct) | 58.2 | - |
| MCTS | 63.8 | 42.1 |
| ReST-MCTS | 65.6 | 36.3 |
| Omega-MCTS | 66.2 | 24.3 |
| CoMCTS | 80.2 | 12.7 |

rate and average search iteration that indicate search effectiveness and efficiency respectively. We can observe that existing tree search methods improve GPT-4o with limited gains. One main reason lies in that traditional MCTS methods generally work by self-bootstrapping and often get trapped in homogeneous low-quality nodes within the reasoning space of a single MLLM. On the other hand, CoMCTS shows great superiority in search effectiveness and efficiency, thanks to the joint expansion mechanism in CoMCTS that allows reasoning-path search not only within the reasoning space of a given MLLM itself but also among those of others, benefiting from the synergy of multiple MLLMs while avoiding being trapped within the reasoning space of a single MLLM.

## 5 Conclusion

This paper presents CoMCTS, a new learning-to-reason approach for MLLMs, which introduces the concept of collective learning into "tree search" for effective and efficient reasoning-path searching and learning. Based on the CoMCTS, we search effective and reflective reasoning paths for a set of multimodal inputs, and construct Mulberry-260k, a multimodal learning-to-reason-and-reflect dataset with a tree of rich, explicit and well-defined reasoning nodes for each question. Using Mulberry-260k, we train our model, Mulberry, a series of MLLMs with o1-like step-by-step Reasoning and Reflection capabilities. Furthermore, extensive experiments, ablation studies and discussion demonstrate the superiority of our proposed methods on various benchmarks. We hope that CoMCTS along with Mulberry-260k and Mulberry will provides valuable resources and offer new insights for multimodal MCTS search and reasoning.

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

# Appendix

We provide theorem statement and assumptions in Section A, implementation details in Section B, more discussions in Section C and additional experiments in Section D. In addition, we discuss the relation to other studies in Section E, provide the analysis of reasoning training data in Section F and qualitative results in Section G. Then, we provide the used prompts in Section H and the sources of the raw data in Section I. In the end, we provide the analysis with error bars in Section J and discussion about the limitations in Section K.

## A   Theorem Statement and Assumptions

**Theorem A.1** (CoMCTS Consistency). *Let $\{\pi_1, \pi_2, \ldots, \pi_K\}$ be a collection of models used in CoMCTS. Suppose the following conditions hold:*

1. *The search space is finite, or after pruning below a threshold $t$, it remains effectively traversable.*

2. *The return $\mu_s$ of each node $s$ lies in a bounded interval $[a, b] \subset \mathbb{R}$.*

3. *Nonzero generation probability and unbiased evaluation:*

   (a) *For each model $\pi_k$, there exists $\epsilon_k > 0$ such that the probability of generating an optimal branch is at least $\epsilon_k$.*

   (b) *The collective evaluation function*

   $$R(s) \;=\; \frac{1}{K} \sum_{l=1}^{K} \pi_l\Big( \cdot \; \Big| \; prompt_{eval}, Q, Parent(s), s\Big)$$

   *has bounded bias, with controllable variance, approximating the true node value $\mu_s$.*

4. *UCB Parameter: In the Selection phase, a UCB strategy with $c > 0$ is used.*

5. *Repeated-Visit Opportunity: All candidate nodes above threshold $t$ can be revisited with positive probability; thus, as $T \to \infty$, each node not pruned is sampled infinitely often if it remains promising.*

*Under these assumptions, as $T \to \infty$, the CoMCTS algorithm visits the optimal path infinitely often, and the value estimate $V(s)$ converges to the true value $\mu_s$ for each node.*

## Proof of Theorem A.1

**Lemma A.2** (UCB Consistency [54]). *In the multi-armed bandit setting with bounded i.i.d. rewards, the UCB1 strategy*

$$\mathrm{UCB}(k) \;=\; \overline{X}_k \;+\; c\sqrt{\frac{\ln n}{n_k}}$$

*ensures that each suboptimal arm is pulled only finitely often, while the optimal arm is sampled infinitely often. Consequently, $\overline{X}_k \to \mu_k$ as $n \to \infty$.*

**Lemma A.3** (Ensemble Near-Unbiasedness and Variance Control). *Suppose $\hat{\mu}_1, \ldots, \hat{\mu}_K$ are unbiased (or bounded-bias) estimators of some $\mu$, each with bounded variance. Then their average remains close to $\mu$ in expectation and can significantly reduce variance compared to a single estimator.*

The proof proceeds by showing four key properties:

1. Optimal children appear in the tree due to nonzero expansion probability ($\epsilon_k > 0$).
2. Optimal nodes are not pruned, via Hoeffding's inequality demonstrating that $R(s^*) > t$ with high probability if $\mu_{s^*} > t$.
3. UCB selection guarantees repeated visits for discovered optimal nodes.
4. Value estimates converge to the true values $\mu_s$ under repeated sampling.

## Optimal Children Appear in the Tree

Let $s^*$ be a child node on the unknown optimal path. Each model $\pi_k$ has probability $\epsilon_k > 0$ of generating $s^*$ during *Expansion*. If these events are approximately independent across $k$, the joint probability of including $s^*$ is

$$p_{\text{expansion}} = 1 - \prod_{k=1}^{K}\left(1 - \epsilon_k\right) > 0.$$

Hence, after enough expansions, $s^*$ almost surely appears in the CoMCTS search tree.

## Optimal Nodes Survive Threshold-Based Pruning

Suppose $s^*$ is truly optimal, with $\mu_{s^*} > t$. In the *Simulation and Error Positioning* step, CoMCTS computes

$$R(s^*) = \frac{1}{K} \sum_{l=1}^{K} \pi_l\left(\cdot \ \Big| \ \text{prompt}_{\text{eval}}, Q, \text{Parent}(s^*), s^*\right).$$

Let each individual score be $X_l$ within a bounded interval. Under near-unbiasedness (Lemma A.3),

$$\mathbb{E}[\,X_l\,] \approx \mu_{s^*}, \quad \mathbb{E}[\,R(s^*)\,] \approx \mu_{s^*}.$$

Define $\delta = \mu_{s^*} - t > 0$. Then

$$\Pr\big[R(s^*) < t\big] = \Pr\Big[\tfrac{1}{K} \sum_{l=1}^{K} X_l < t\Big] = \Pr\Big[\sum_{l=1}^{K} X_l - K\,\mu_{s^*} \leq -K\,\delta\Big].$$

By Hoeffding's inequality, for $X_l$ and approximate independence, there holds

$$\Pr\Big[\sum_{l=1}^{K} X_l < K\,t\Big] \leq \exp\big(-K\,\delta^2\big).$$

Therefore, a truly optimal node $s^*$ with $\mu_{s^*} > t$ is pruned with exponentially small probability, implying $s^*$ remains in the tree almost surely.

## UCB Selection Guarantees Repeated Visits

Nodes not pruned enter the *Selection* stage, using

$$\text{UCB}(s) = V(s) + c\sqrt{\frac{\ln\big(N(\hat{s})\big)}{1 + N(s)}},$$

where $\hat{s}$ is the parent of $s$. By analogy to multi-armed bandits (Lemma A.2), if $s^*$ is the highest-value child, it gets chosen infinitely often. Hence $N(s^*) \to \infty$ as $T \to \infty$.

## Value Estimates Converge

When a node $s$ returns score $R(s)$, backpropagation updates:

$$V(\hat{s}) \leftarrow \frac{N(\hat{s})\,V(\hat{s}) + \sum_{s_l \in \text{Child}(\hat{s})} R(s_l)}{N(\hat{s}) + \text{CountChild}\big(S^*_{\text{candidate}}, \hat{s}\big)},$$

$$N(\hat{s}) \leftarrow N(\hat{s}) + \text{CountChild}\big(S^*_{\text{candidate}}, \hat{s}\big).$$

As $N(s^*) \to \infty$, the empirical average $V(s^*)$ converges to $\mu_{s^*}$ by standard arguments (e.g. the law of large numbers for bounded returns). Ancestor nodes on the optimal path also receive increasingly accurate updates, ensuring $V(s) \to \mu_s$.

Combining all above results, we conclude that CoMCTS inherits the same consistency guarantees as standard MCTS, with optimal paths ultimately visited infinitely often and $V(s) \to \mu_s$.

# B  Training Details

## B.1  Implementation Details

**CoMCTS implementation Details.** We implement collective learning in CoMCTS with four models, including GPT-4o, Qwen2-VL-7B, LLaMA-3.2-11B-Vision-Instruct, and Qwen2-VL-72B, to construct Mulberry-260K. In CoMCTS, we set maximum search iteration to 20 and threshold $t$ in Eq. 3 to 0. In each iteration, each model generates one candidate reasoning path to balance search exploration and exploitation. In CoMCTS, we set the temperature to 0.9 for all models to balance creativity and determinism. For all other hyperparameters, we use the same default values as in the respective base models.

**Model training details.** In this paper, we use Mulberry-260K data searched by CoMCTS to train the Mulberry series models based on four popular MLLMs, including both the search models included in CoMCTS (*i.e.*, Qwen2-VL-7B and LLaMA-3.2-11B-Vision-Instruct) and the models not within the search (*i.e.*, Qwen2-VL-2B and LLaVA-NeXT-8B). We use the LLaMA-factory training framework [55] to train our models, using 8 NVIDIA H100 GPUs.

For the training scheme, all models are optimized for 2 epoch with the AdamW optimizer and a cosine learning schedule. We fine-tune the models with a batch size of 128 and employ DeepSpeed ZeRO-3 strategy to optimize memory consumption. A unified learning rate of 1e-5 is used for LLaVA-NeXT-8B and LLaMA-3.2-11B-Vision-Instruct, while for Qwen2-VL-2B and Qwen2-VL-7B, the learning rates are adjusted to 2e-5 and 5e-6, respectively. As the number of tokens in the responses is significantly greater than that of direct simple answers, the training time increases accordingly, as detailed in Tab. 5.

## B.2  Training Loss Curve

We provide more in-depth training details regarding the training loss curve in Fig. 4. Using the Mulberry-260K dataset for collective supervised fine-tuning, consistent and stable training is achieved across all four models, with the training loss steadily decreasing, demonstrating the training stability of Mulberry-260K searched by CoMCTS.

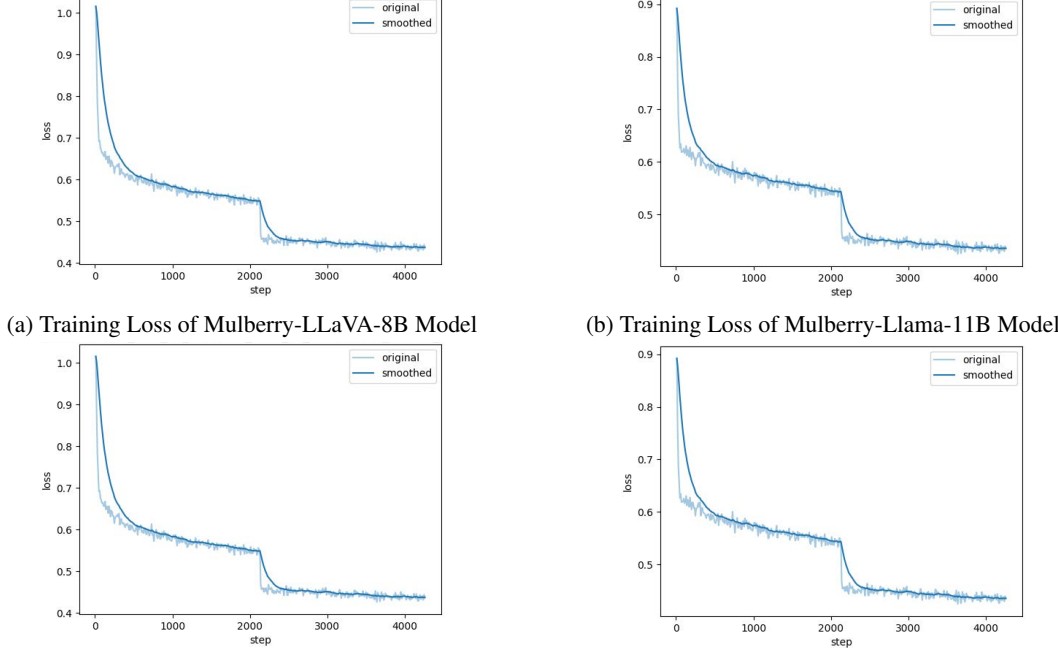

(a) Training Loss of Mulberry-LLaVA-8B Model

(b) Training Loss of Mulberry-Llama-11B Model

(c) Training Loss of Mulberry-2B Model

(d) Training Loss of Mulberry-7B Model

Figure 4: Training loss curve.

Table 5: **Training time.** We provide the training time for Mulberry series models trained on Mulberry-260K data generated by CoMCTS. The experiments are conducted on 8 NVIDIA H100 GPUs, using Llama-Factory framework [55].

| Metrics | Mulberry-LLaVA-8B | Mulberry-Llama-11B | Mulberry-2B | Mulberry-7B |
|---|---|---|---|---|
| Training Time (hours) | 12.5 | 11.2 | 4.5 | 6.0 |

# C   More Discussions

## C.1   Detailed Comparison with the Existing COT Training Dataset

We provide detailed comparison of Mulberry-260K with the existing COT training dataset (i.e., LLaVA-Reasoner and LLaVA-CoT-100K) in Table 6. Table 6 compares key attributes such as dataset size (number of images and QA pairs), average token length of questions and responses, and the inclusion of reasoning and reflection. As shown, Mulberry-260K not only includes both reasoning and reflection, but also provides longer and more informative responses, underscoring its value as a CoT training resource.

Specifically, unlike LLaVA-Reasoner, which generates CoT reasoning by directly prompting GPT-4o, Mulberry-260K employs the proposed CoMCTS search, producing step-by-step reasoning paths that are not only longer but also feature more precise intermediate steps, benefiting from collective expansion and error positioning in CoMCTS.

Different to LLaVA-CoT, which uses structured reasoning with a fixed flow of certain pre-defined stages, Mulberry-260K leverages the proposed CoMCTS to dynamically search for both reasoning and reflection data, resulting in more detailed and logically coherent intermediate steps.

Furthermore, thanks to CoMCTS, Mulberry-260K is the first multimodal step-by-step reasoning dataset equipped with reflective capabilities, fostering deeper exploration of reflection in MLLMs.

Table 6: Detailed comparison with the existing COT training dataset.

| Datasets | Images | QA-Pairs | Avg Question token | Avg Response token | Reasoning | Reflection |
|---|---|---|---|---|---|---|
| LLaVA-Reasoner | ∼404K | ∼404K | 31.4 | 122.0 | Y | N |
| LLaVA-CoT-100K | ∼100K | ∼255K | 23.2 | 212.4 | Y | N |
| Mulberry-260K | ∼260K | ∼260K | 31.9 | 234.5 | Y | Y |

## C.2   Correctness of the Generated Reasoning Paths

As discussed in Limitation, neither our method nor advanced reasoning models (e.g., o1, QvQ, DeepSeek-R1) can ensure every step is 100% correct. Similar limitations are noted in QvQ. Here, we provide the manual evaluation of 100 reasoning paths. As shown in Table 7, our CoMCTS yields a much lower error rate in reasoning paths compared to MCTS.

Table 7: Manual evaluation of the correctness of the CoMCTS-generated reasoning paths.

| Methods | Reasoning Path Error Rate |
|---|---|
| MCTS | 42% |
| CoMCTS | 18% |

## C.3   Training MLLMs on CoMCTS vs. Other Tree Search Data.

We examine the quality of the CoMCTS generated data by providing the experiments using data generated from standard MCTS. As shown in Table 8, models are trained on 26K samples searched by MCTS and CoMCTS, respectively. By incorporating collective knowledge, CoMCTS generates more accurate reasoning paths, leading to a 2.1% performance improvement.

Table 8: Training MLLMs on CoMCTS vs. other Tree Search data.

| Methods | MathVista |
|---|---|
| LLaMA-V-11B | 48.6 |
| MCTS | 50.3 |
| CoMCTS | 52.4 |

# D  Additional Experiments

In this section, we include additional experiments on CoMCTS and Mulberry models. Unless otherwise specified, the experiments in this section are based on the Qwen2-VL [2] model.

## D.1  More Detailed Ablation Study on CoMCTS

We provide a more detailed analysis of search success rate in the CoMCTS ablation study in Table 9, studying the direct predictions of search models and their performance in CoMCTS. We can observe that due to the model's limited reasoning capability, directly performing reasoning prediction with a single model yields poor performance. When using only Qwen2-VL-7B, the success rate of direct predictions is just 37.4%, and even the powerful closed-source model GPT-4o achieves only 58.2% accuracy in direct reasoning predictions. However, by leveraging tree search and collective knowledge, our proposed CoMCTS significantly improves the search success rate to 80.2% with four search models, demonstrating the effectiveness of CoMCTS.

Table 9: **Ablation Study on CoMCTS.** We study how each model in CoMCTS collective learning contribute to overall tree search performance in Search Success Rate (S.S.R.).

| Direct Prediction | | | | CoMCTS | | | | S.S.R. |
|---|---|---|---|---|---|---|---|---|
| GPT-4o | Qwen2-VL-7B | Llama3.2-11B | Qwen2-VL-72B | GPT-4o | Qwen2-VL-7B | Llama3.2-11B | Qwen2-VL-72B | |
| ✔ | | | | | | | | 58.2 |
| | ✔ | | | | | | | 37.4 |
| | | ✔ | | | | | | 35.4 |
| | | | ✔ | | | | | 48.3 |
| | | | | ✔ | | | | 63.8 |
| | | | | ✔ | ✔ | | | 66.2 |
| | | | | ✔ | ✔ | ✔ | | 69.7 |
| | | | | ✔ | ✔ | ✔ | ✔ | 80.2 |

## D.2  Parameter Studies

**Threshold $t$.** We conduct a parameter study on the threshold $t$ of Equation 3 in CoMCTS, analyzing its impact on search success rate and average search iterations, as shown in Table 10. In the CoMCTS tree, the node value $V$ represents the quality of this reasoning step, and the threshold $t$ determines whether the node is removed based on its value, thereby regulating the overall quality of nodes. Setting $t$ to a higher value, *i.e.*, $t = 0.2$, enhances the quality at each step but also slows down the expansion of nodes. As the quality of each node improves, search models are more likely to find the correct path within the collective search space, leading to an increase in search success rate. However, as node expansion slows, the average search iterations increase significantly. Setting $t$ to a smaller value, *i.e.*, $t = -0.2$, increases the likelihood of nodes containing irrelevant information, disrupting the reasoning process. This results in a slight decrease in search success rate and a modest increase in average search iterations. On the other hand, setting the threshold as zero, *i.e.*, $t = 0$, achieves a great trade-off in search success rate and average search iterations, balancing search effectiveness and efficiency.

**Temperature and repetition penalty.** As shown in Tables 11-12, we also conduct extra hyperparameter studies (e.g., temperature and repetition penalty), indicating our method is tolerant to these parameters.

Table 10: **Parameter Study of Threshold** $t$. We study the impact of different threshold values $t$ in Equation 3 of CoMCTS on the search success rate and average search iterations.

| Threshold | Search Success Rate | Average Search Iteration |
|---|---|---|
| -0.2 | 79.8 | 12.9 |
| 0 | 80.2 | 12.7 |
| 0.2 | 80.9 | 16.3 |

Table 11: **Parameter Study of Temperature.**

| Temperature | Search Success Rate |
|---|---|
| 0.7 | 80 |
| 0.8 | 79 |
| 0.9 (we use) | 81 |
| 1.0 | 80 |
| 1.1 | 78 |

### D.3 Discussion on Step Separators

We conduct experiments to discuss the impact of different step separators in Table 13. First, the absence of step separators results in disorganized reasoning logic, leading to a notable decline in model performance, with an accuracy of only 48.4% on MathVista. This result highlights the importance of step separators in reasoning, as they enhance MLLMs' ability to comprehend the logical structure within reasoning paths. Next, we compare the special tokens separators (*e.g.*, `<step_1>`, `<step_2>`, etc) with context separators (*e.g.*, ###, as illustrated in Figure 7). The former requires adding special tokens to the vocabulary, while the latter relies on predefined formatting rules. The results in Table 13 show that the context separators method outperform the special token separators, achieving an accuracy of 51.7% compared to 50.6%. This performance difference may stem from models' inherent tendency to use markdown formatting (*e.g.*, ###) to highlight answers and separate steps, which could lower the cognitive associated with learning this pattern in limited data.

### D.4 Discussing the Importance of Reasoning

We discuss the importance of step-by-step reasoning in Tab. 14. Specifically, using the same 260K multimodal data, we compare the results trained with direct answers to those trained with step-by-step reasoning searched by CoMCTS. Experimental results show that training with reasoning responses can substantially enhance model performance. For example, on MathVista, it outperforms direct answer training by a margin of 4.9%. This improvement demonstrates the potential of reasoning to enhance MLLM performance.

### D.5 Training with Different Proportions of Mulberry-260K

As shown in Tab. 15, we conduct the suggested studies by training models using different proportions of Mulberry-260K data over the base model LLaMA-3.2-Vision-11B-Instruct. We can observe that Mulberry consistently improves performance across all Mulberry-260K data proportions, demonstrating its effectiveness in limited-data scenarios. Besides, the performance gains increase steadily as more Mulberry-260K data is used, highlighting its strong scalability.

### D.6 Comparison with Base Models Trained using Other COT Data

As shown in Table 16, we conduct the comparison for analyzing the impact of different CoT training datasets on the same base model. Using the LLaMA3-LLaVA-NeXT-8B base model, despite using less training data, our Mulberry-LLaVA-8B outperforms LLaVA-Reasoner-8B by +3.1% in average performance, which is largely because LLaVA-Reasoner learns reasoning responses directly distilled from GPT-4o while our Mulberry learns from more effective reasoning data searched by the proposed CoMCTS.

Similarly, with the LLaMA-3.2-Vision-11B-Instruct base model, our Mulberry-LLaMA-11B achieves better results using only 100K QA pairs, significantly fewer than the 255K QA pairs used by

Table 12: **Parameter Study of Repetition Penalty.**

| Repetition penalty | Search Success Rate |
|---|---|
| 0.8 | 78 |
| 0.9 | 79 |
| 1.0 (we use) | 81 |
| 1.1 | 79 |
| 1.2 | 76 |

Table 13: **Discussion on Step Separators.** We study the impact of different reasoning step separators on model performance.

| Methods | MathVista Accuracy |
|---|---|
| w/o Separators | 48.4 |
| Special Token Separators | 50.6 |
| Context Separators | 51.7 |

Table 14: **Discussing the Importance of Reasoning.** We compared the results of models trained on direct answer data with those trained on Mulberry-260K reasoning data.

| Benchmark | Baseline | Direct Answers | Step-by-step Reasoning |
|---|---|---|---|
| MathVista | 43.0 | 46.8 | 51.7 |

LLaVA-CoT-11B. When scaling our dataset to 260K QA pairs, Mulberry-LLaMA-11B outperforms LLaVA-CoT-11B by +2.7% on average, further demonstrating the advantages of CoMCTS-searched reasoning paths over the pre-defined structured reasoning methods in LLaVA-CoT.

# E   Relation to Other Studies

## E.1   Relation to Other MCTS Methods

We summarize the contributions of CoMCTS and highlight the differences compared to previous MCTS methods [24, 9, 7]: (a) This paper presents CoMCTS, a new learning-to-reason approach for MLLMs, which introduces the concept of collective learning into "tree search" for effective and efficient reasoning-path searching and learning. (b) The joint expansion mechanism enables CoMCTS to concatenate reasoning trajectories from multiple MLLMs via iterative search, ultimately constructing a unified reasoning tree comprising diverse and complementary reasoning nodes. This allows reasoning-path searches across multiple models, leveraging their synergy while avoiding traps in low-quality, homogeneous nodes within a single MLLM's reasoning space. (c) In each search iteration, CoMCTS skips multiple intermediate steps and selects the last correct step as the next start node in the joint simulation and error positioning operation, largely reducing search time while maintaining search effectiveness. Additionally, collective knowledge enhances error positioning, allowing models to better identify errors through mutual validation.

## E.2   Relation to Other Reasoning MLLMs

Multimodal large language models [56, 57, 58, 59, 11, 60, 61] have made remarkable progress, with recent MLLMs increasingly focusing on reasoning and intermediate steps [62, 63, 4, 3, 49]. We summarize the contributions of our models and highlight the differences compared to previous reasoning MLLMs: (a) Based on the CoMCTS, we search effective and reflective reasoning paths for multimodal inputs, and construct Mulberry-260k, a multimodal reason and reflect dataset with a tree of rich, explicit and well-defined reasoning nodes for each question. Mulberry-260k is then used to train reasoning model Mulberry. (b) Unlike pre-defined stage-based methods, CoMCTS searches for step-level reasoning data of varying lengths. Training Mulberry models on these CoMCTS-searched reasoning paths equips it with the ability to perform flexible step-by-step reasoning. (c) To the best of our knowledge, Mulberry is the first work to explore the reflective capabilities of MLLMs through tree search and collective knowledge.

Table 15: Training with different proportions of Mulberry-260K.

| Models | Percentage of Mulberry-260K | QA-Pairs | MathVista | MMStar | MMMU | AVG |
|---|---|---|---|---|---|---|
| LLaMA-3.2-V-11B (Baseline) | - | - | 48.6 | 49.8 | 41.7 | 46.7 |
| Mulberry-11B | 10% | ∼26k | 52.4 | 53.1 | 42.8 | 49.4 |
| | 30% | ∼78k | 54.0 | 56.1 | 43.6 | 51.2 |
| | 50% | ∼130k | 58.1 | 57.9 | 44.9 | 53.6 |
| | 100% | ∼260k | 61.1 | 58.5 | 45.6 | 55.1 |

Table 16: Comparison of training with different CoT datasets on the same base model.

| Models | Base Model | CoT Data | QA-Pairs | MathVista | MMStar | MMMU | AVG |
|---|---|---|---|---|---|---|---|
| LLaMA3-LLaVA-NeXT-8B | - | - | - | 37.5 | 42.1 | 41.7 | 40.4 |
| LLaVA-Reasoner-8B | LLaMA3-LLaVA-NeXT-8B | LLaVA-Reasoner-404K | ∼404K | 50.6 | 54.0 | 40.0 | 48.2 (+7.8) |
| Mulberry-LLaVA-8B | LLaMA3-LLaVA-NeXT-8B | Mulberry-260K | ∼260K | 56.3 | 54.5 | 43.0 | 51.3 (+10.9) |
| LLaMA-3.2-Vision-11B-Instruct | - | - | - | 48.6 | 49.8 | 41.7 | 46.7 |
| LLaVA-CoT-11B | LLaMA-3.2-Vision-11B-Instruct | LLaVA-CoT-100K | ∼255K | 54.8 | 57.6 | 44.9 | 52.4 (+5.7) |
| Mulberry-Llama-11B | LLaMA-3.2-Vision-11B-Instruct | Mulberry-100K | ∼100K | 56.8 | 57.8 | 44.7 | 53.1 (+6.4) |
| Mulberry-Llama-11B | LLaMA-3.2-Vision-11B-Instruct | Mulberry-260K | ∼260K | 61.1 | 58.5 | 45.6 | 55.1 (+8.4) |

# F  Analysis of Reasoning Training Data

## F.1  Analysis of Reasoning Data Distribution.

We analyze the CoMCTS-searched reasoning paths in Mulberry-260K by examining the distribution of reasoning steps, as shown in Figure 5. Figure 5 shows that reasoning steps predominantly fall between 6 and 8, with an average of 7.5 across the entire Mulberry-260k. For simple reasoning tasks, the chart-related subset of Mulberry-260k, reasoning steps typically ranges from 6 to 7, averaging 6.8. In contrast, for complex mathematical and logical reasoning tasks, such as the geometry-related subset of Mulberry-260k, the distribution shifts to 7 and 10 steps, with an average of 8.9. These observations highlight that CoMCTS's collective tree search design generates effective reasoning trajectories with flexible numbers of steps, learning from which allows to train a powerful MLLM with reasoning flexibility, *i.e.*, a model can "think less and faster" when handling simple questions (*i.e.*, allocate and generate fewer intermediate reasoning steps) and "think more and slower" when tackling complex tasks (*i.e.*, allocate and generate a greater number of intermediate reasoning steps).

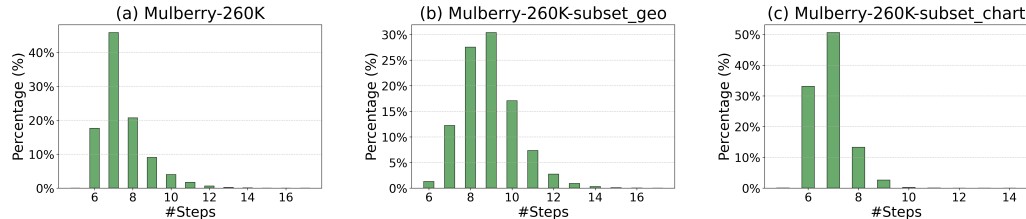

Figure 5: Distribution of reasoning steps in Mulberry-260K data.

## F.2  Analysis of Effective Reasoning Training Data

We provide an example of effective reasoning data generated by CoMCTS in Figure 6. In this reasoning path, an accurate image analysis is first searched, along with reasonable rationales, allowing CoMCTS to subsequently search logical and correct step-by-step reasoning based on precise perception and rationales, ultimately deriving the correct answer. This clear reasoning largely due to CoMCTS's collective expansion and collective error positioning, allowing search models to complement each other, refine error identification, and iteratively improve reasoning paths.

## F.3  Analysis of Reflective Reasoning Training Data

We provide an example of reflective training data generated by CoMCTS in Figure 7. In this reasoning path, one search model identifies correct reasoning path after several iterations, while CoMCTS's

collective error positioning finds a perception error at Step 1 in another path. CoMCTS then integrates the incorrect Step 1 into the correct reasoning path (*i.e.*, before its sibling nodes) with a reflective prompt to create reflective data. Here, incorrect Step 1 misjudges the proportion of "Medium Orchid", but later steps identify and correct the error. Fine-tuning with this reflective data equips the model with the ability to reassess previous steps when generating the next step, leading to more accurate reasoning and improves performance.

# G    Qualitative Results

## G.1    Qualitative Comparison

We provide a qualitative comparison of LLaVA-NeXT-8B [50], Qwen2-VL-7B [2], and Mulberry-7B in Figure 8.  LLaVA-NeXT-8B and Qwen2-VL-7B generate relatively short predictions without thorough thinking, leading to incorrect answers. In contrast, our Mulberry, trained with CoMCTS-searched reasoning data, creates rich, explicit and well-defined reasoning steps with comprehensive understanding, ultimately arriving at the correct answer.

## G.2    Reflective Qualitative Results

We present a reflective qualitative results of Mulberry in Figure 9. Fine-tuning on reflective reasoning data, Mulberry models develop the capability to reflect on previously generated step. Here, Mulberry initially misidentifies the accuracy of the parent but, through continuous reflection, detects the error at Step 2, corrects it, and ultimately arrives at the correct answer.

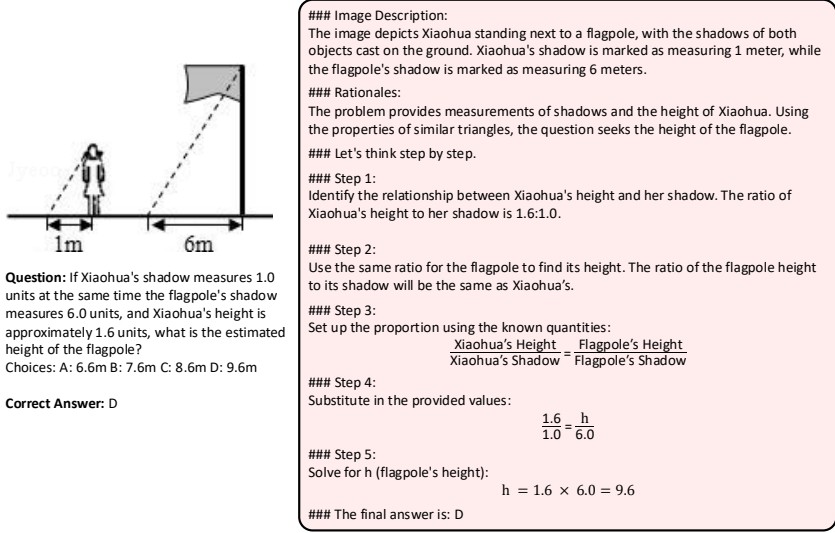

Figure 6: **An example of effective reasoning training data.**

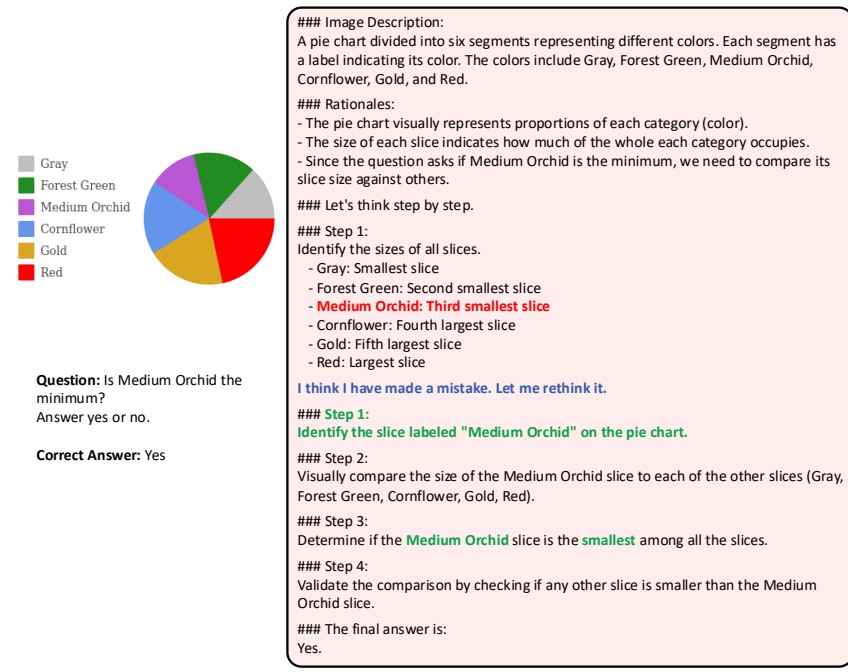

Figure 7: **An example of reflective reasoning training data.**

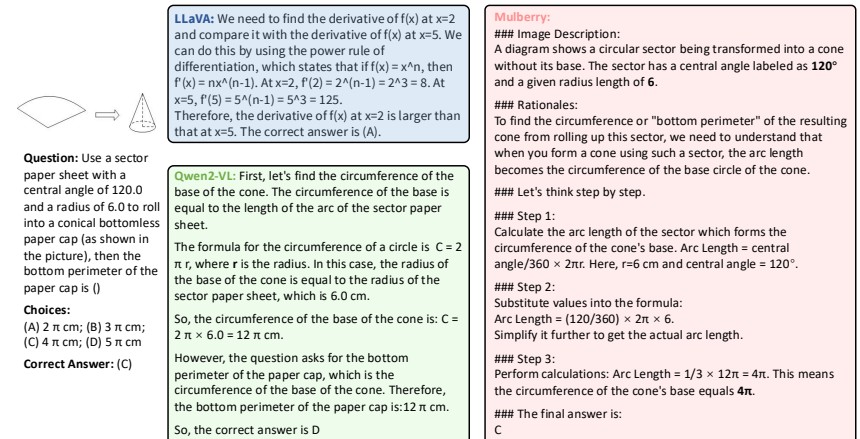

Figure 8: **Qualitative Comparison.** Our Mulberry, trained with CoMCTS-searched reasoning data, creates rich, explicit and well-defined reasoning steps with comprehensive understanding, ultimately arriving at the correct answer.

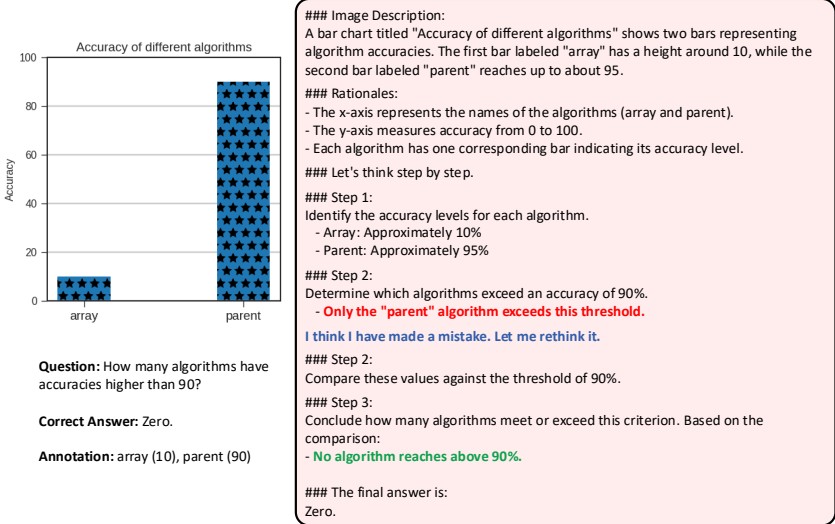

Figure 9: **Reflective Qualitative Results.** Our Mulberry, trained with CoMCTS-searched reflection data, generated reflective reasoning, ultimately arriving at the correct answer.

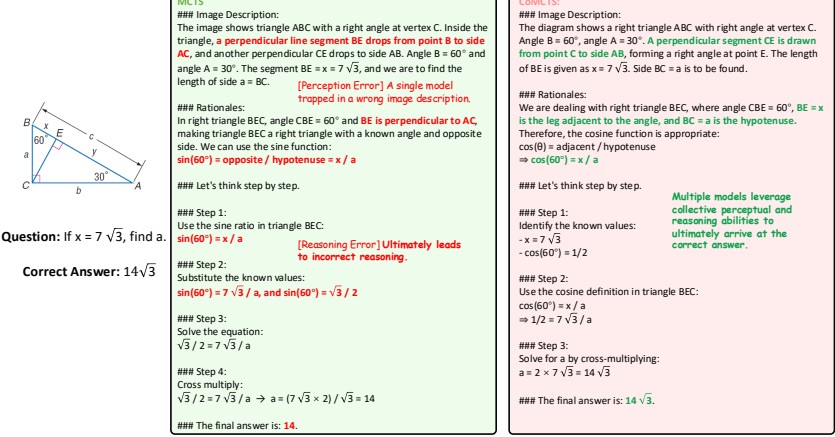

Figure 10: **An example to compare CoMCTS with other tree search methods.**

### G.3 Examples to Compare CoMCTS with Other Tree Search Methods

We provided a qualitative comparison between the CoMCTS and single model MCTS.

When a single model performs step-by-step reasoning, an error in the initial image understanding can lead to a cascade of incorrect reasoning steps, ultimately resulting in a wrong answer. However, when multiple models are used collaboratively for reasoning, if one model fails to correctly interpret the image, others can help identify the correct steps. By cross-verifying and complementing each other's interpretations, the ensemble can jointly search for and arrive at the correct answer.

As illustrated in Figure 10, the single-model MCTS failed to correctly interpret the image and got trapped in an erroneous node, leading to a reasoning path that consistently failed to arrive at the correct answer.

## H   Prompt

In this section, we present the prompts involved in different operations of CoMCTS to execute various task instructions. For expansion in operation (a), we use the prompt in Figure 11 and Figure 12. The

prompt in Figure 11 is used to expand the root node for the closed-source model GPT-4o and all nodes for the open-source models. It is worth noting that, since open-source models can directly and forcefully incorporate preceding reasoning steps, allowing the model to generate subsequent steps based on previous responses, whereas closed-source models cannot. Therefore, we design specific prompts for closed-source models to expand non-root nodes. For GPT-4o, we use the prompt shown in Figure 12 to expand non-root nodes in the existing search tree and generate subsequent reasoning steps. For error positioning in operation (b), we use the prompt shown in Figure 13 to evaluate the correctness of each step. Each step is scored as correct, neutral, or incorrect, with scores of 1, 0, and -1, respectively. The evaluation results for each model are then combined through a weighted average. Finally, we use the prompt in Figure 14 to verify if the final answer from reasoning path matches the ground truth.

```
Generate an image description based on the question.
Then, provide a rationale to analyze the question.
Next, generate a step-by-step reasoning process to solve the problem. Ensure the steps are logical
and concise.
Finally, provide a concise summary of the final answer in the following format: 'The final answer is:
xxx'. If the question is multiple-choice, provide the options along with their content. If it is free-
form, directly present the final result. Do not provide any explanation.

Format your response with the following sections, separated by ###:
### Image Description:
### Rationales:
### Let's think step by step.
### Step 1:
### Step 2:
...
### The final answer is:

{QUESTION}
```

Figure 11: **The prompt for expansion.** The prompt expands root nodes in closed-source and all nodes in open-source models.

```
Generate an image description based on the question.
Then, provide a rationale to analyze the question.
Next, generate a step-by-step reasoning process to solve the problem. Ensure the steps are logical
and concise.
Finally, provide a concise summary of the final answer in the following format: 'The final answer is:
xxx'. If the question is multiple-choice, provide the options along with their content. If it is free-
form, directly present the final result. Do not provide any explanation.

Format your response with the following sections, separated by ###:
### Image Description:
### Rationales:
### Let's think step by step.
### Step 1:
### Step 2:
...
### The final answer is:

{QUESTION}

Please complete the response based on the reasoning prefix without altering its content.

Reasoning prefix: {REASONING_PREFIX}
```

Figure 12: **The prompt for expansion.** The prompt expands non-root nodes in closed-source models.

# I    The Sources of Raw Data

To construct a comprehensive and general-purpose tree-based reasoning dataset, we collect 260K raw multimodal input questions spanning various domain, including

- 55K Mathematical Data: From GLLaVA [64], GEOS [65], UniGeo [66], GeoQA Plus [53], Geo3K [52], MathVision [67], GeoMverse [68], and MathV360K [69]. These datasets cover a broad spectrum of mathematical problems, where solving requires extensive reasoning

```
### Question:
{QUESTION}

### Ground truth answer:
{GT_ANSWER}

### Reasoning steps:
{REASONING}

Given the question and reasoning steps listed above, along with the corresponding ground truth
answer, please evaluate the correctness of the image description, rationales, and each step of the
reasoning process.

Requirements:
1. Output the decision ("correct", "neutral", "incorrect") for each step following the format of
"Final Decision:\nImage Description: [your decision]; Rationales: [your decision]; Let's think step
by step: [your decision]; Step 1: [your decision]; Step 2: [your decision]; ...";
2. Do not provide any explanation.
```

Figure 13: **The prompt for error positioning.**

```
Evaluate whether the model's answer matches the correct result.

- If it does not align, respond with 'No'.
- If the model's answer aligns with the correct result, respond with 'Yes'.

Provide only 'Yes' or 'No' as the output, with no explanation.

The question is: {QUESTION}

The model's answer is: {MODEL_ANSWER}

The correct result is: {GT_ANSWER}
```

Figure 14: **The prompt for evaluating the final result.**

and multiple logical steps, highlighting their value in Mulberry-260K. We apply CoMCTS
search to the raw data from these datasets to generate reasoning paths, which are then used to
train Mulberry models, equipping them with advanced mathematical and logical reasoning
skills.

- 116K Figure Understanding data: From DVQA [70], DocVQA [71], FigureQA [72],
  PlotQA [73], ChartQA [74], InfoVQA [75], MultiHiertt [76], and LRV-Chart [77]. These
  datasets cover various figure types and understanding tasks, including charts, bar graphs,
  pie charts, histograms, etc. Training on these data searched by CoMCTS equips the model
  with figure reasoning capabilities such as table computation, information retrieval, and trend
  analysis.

- 41K Math Word Problem Data: From IconQA [78], TabMWP [79], CLEVR [80], CLEVR-
  Math [81], and Super-CLEVR [82]. These datasets span various mathematical word reason-
  ing tasks. Training on them in Mulberry-260K strengthens the model's counting, arithmetic,
  and ogical deduction skills, enhancing its accuracy and interpretability in solving complex
  math word problems.

- 2K Medical Data: From VQA-RAD [83], and PMC-VQA [84]. These datasets focus on
  medical visual understanding and reasoning, encompassing a variety of radiology images
  and diseases. Leveraging this subset enhances the model's capability in medical image
  comprehension and diagnostic reasoning.

- 17K Science Data: From TQA [85], AI2D [86], and ScienceQA [87]. These datasets in
  CoMCTS plays a critical role in enhancing the model's ability to tackle complex scientific
  problems, perform multi-modal reasoning, and interpret scientific illustrations and textual
  descriptions cohesively.

- 24K Nature World QA Data: From VQA-AS [88], A-OKVQA [89], TextVQA [90],
  Vizwiz [91], and VQA2.0 [92]. These datasets encompass tasks involving naturalscenes,
  textual elements, and open-ended visual question answering, challenging models to interpret
  complex visuals, understand embedded text, and generate accurate responses. This subset

plays a key role in enhancing the model's ability to reason about real-world visual and textual content across diverse contexts.

## J Error Bars

We provide the error bars of different Mulberry models on the MathVista benchmark, as shown in Tab. 17. We conduct five repeated evaluations to obtain the error bars, and the results indicate that the error bars fall within a narrow range.

Table 17: **Error Bars.** The experimental results are based on five repeated evaluations.

| Benchmark | Mulberry-LLaVA-8B | Mulberry-Llama-11B | Mulberry-2B | Mulberry-7B |
|---|---|---|---|---|
| MathVista | $56.3 \pm 0.3$ | $61.1 \pm 0.2$ | $51.7 \pm 0.3$ | $63.1 \pm 0.2$ |

## K Limitations

Mulberry is a preliminary exploration work in o1-like MLLM, leveraging Collective Monte Carlo Tree Search to enable effective and efficient reasoning-path searching and learning. CoMCTS leverages collective knowledge to significantly improve the search success rate and efficiency of reasoning path search. By training on the reasoning data generated through CoMCTS, Mulberry has gained step-by-step reasoning capabilities, leading to a substantial improvement in overall performance. Nevertheless, certain limitations must be acknowledged.

**Hallucinations in intermediate steps.** Hallucinations are still prevalent in MLLMs, whether in closed or open-source models. For instance, the models may generate obvious errors in intermediate reasoning steps yet still arrive at the correct final answer in CoMCTS. Therefore, although we incorporated multiple models to better detect errors, some errors still persist in the intermediate steps because ensuring the correctness of all intermediate steps often requires human checks, which is extremely costly and unaffordable for us.

**Error localization.** During our experiments, we observed that models struggle to detect their own errors. To address this, CoMCTS employs multiple models to cross-check each other's errors. However, our findings also revealed that smaller models often fail to generate effective detection responses, while larger models occasionally exhibit inaccurate error localization. Thus, inaccurate localization may impact the efficiency of the search and we recommend using larger models for error localization or exploring better prompts to enable smaller models to localize errors more accurately.

**Reflective capability.** Mulberry is an early-stage work in exploring reflective capabilities within the field of MLLMs. We would like to clarify that since reflective data constitutes only a small portion of the entire dataset, the Mulberry model generates reflective responses occasionally. We hope to further explore the model's reflective capabilities in future work.

