# OpenReview forum: "Mulberry: Empowering MLLM with o1-like Reasoning and Reflection via Collective Monte Carlo Tree Search"
_NeurIPS.cc/2025/Conference — NeurIPS 2025 spotlight_

### Official Review · Reviewer_YpLW · 2025-06-23

**Clarity:** 3
**Significance:** 3
**Originality:** 2
**Rating:** 4
**Confidence:** 4

**Summary:**

This paper introduces Mulberry, a multimodal large language model (MLLM) enhanced with step-by-step reasoning and reflection capabilities, achieved through the Collective Monte Carlo Tree Search (CoMCTS) method. CoMCTS leverages collective knowledge from multiple models to collaboratively search, evaluate, and refine reasoning paths, addressing challenges like search effectiveness and efficiency in traditional MCTS. This paper also constructs a dataset with rich reasoning trees and uses it to train Mulberry, which outperforms existing models on benchmarks requiring complex reasoning.

**Questions:**

1. What are the innovative aspects of the CoMCTS tree search method designed for MLLMs, and how does it differ from text-based methods?

2. How can the overhead introduced by CoMCTS during the search phase be measured? If feasible, could you estimate the total token cost, GPU hours, or API price for data generation?

3. Could you provide statistics on the contribution of each model in the *Collective* process, such as the similarity in sampling between different models or the proportion of each model in generating correct paths?

4. Is it possible to achieve good performance without introducing smaller models, for example, by only using GPT-4o and Qwen2-VL-72B? For Qwen2-VL-72B and Qwen2-VL-7B, their behaviors might be similar due to shared training data—does integrating both models still provide an advantage?

**Ethical Concerns:**

["NO or VERY MINOR ethics concerns only"]

**Final Justification:**

After reviewing the author's rebuttal, several questions have been addressed, prompting me to increase my overall rating.

**Limitations:**

yes

**Quality:**

2

**Strengths And Weaknesses:**

## Strengths

1. The paper introduces Collective Monte Carlo Tree Search (CoMCTS), a novel approach that leverages collective knowledge from multiple models to enhance reasoning-path searching and learning in MLLMs.

2. The construction of Mulberry-260k, a multimodal dataset with rich, explicit reasoning nodes, provides a valuable resource for advancing research in step-by-step reasoning and reflection.

3. The paper demonstrates strong performance of the Mulberry models across various benchmarks, outperforming other models.

## Weaknesses

1. The CoMCTS algorithm merely integrates model ensemble techniques with the MCTS algorithm, lacking significant novelty. Previous work, like LE-MCTS [1], has already investigated this combination in NLP tasks. Furthermore, CoMCTS does not exhibit a specialized design for multimodal models. For example, *expansion, simulation and error positioning, backpropagation, and selection* can also be applied to standard LLMs, and some related research on these approaches already exists.

2. The performance improvement from MCTS alone appears marginal. For instance, in Table 2, GPT-4o direct achieves 58.2, while GPT-4o + CoMCTS reaches only 63.8—a modest gain given the increased computational overhead. The more significant improvement seems to stem from ensembling the Qwen2-VL-72B model rather than MCTS itself.

3. The effectiveness of CoMCTS hinges on the diversity and quality of the models in the collective learning group. Additionally, the reliance on multiple models for collective learning and iterative tree search increases the computational overhead.

[1] Park, Sungjin et al. “Ensembling Large Language Models with Process Reward-Guided Tree Search for Better Complex Reasoning.” North American Chapter of the Association for Computational Linguistics (2024).

---

> ### Author Rebuttal · Authors · 2025-07-31
>
> Dear reviewer YpLW,
>
> Many thanks for your professional, detailed, and valuable reviews.
> We are encouraged by your recognition of CoMCTS as a novel approach, the valuable resource of the Mulberry-260K dataset, and the strong performance of Mulberry models.
> We will do our best to address your concerns one by one.
>
> ---
> **W1: Discussion of the Uniqueness and Distinction of CoMCTS**
>
> Thank you for raising this concern.
> We respectfully believe that LE-MCTS and our paper are concurrent works (completed within the same week). Given its relevance and excellence, we will include a discussion of the differences between our work and LE-MCTS, and cite it appropriately in our revised paper.
> Below, we summarize the key distinctions between CoMCTS and LE-MCTS:
> 1) Different objectives. Our work uses CoMCTS to generate a novel, effective, and reflective reasoning dataset for MLLM training, whereas LE-MCTS focuses solely on inference for LLMs.
> 2) New dataset constructed by CoMCTS: We use CoMCTS to build a large-scale reasoning and reflection dataset, which we hope will benefit and facilitate further research in the community.
> 3) Different MCTS design. In each iteration, our model generates a complete reasoning path, followed by pruning error nodes, as detailed in Lines 146-162. We believe this approach is more suitable for LLM/MLLM compared to traditional step-by-step generation, making data generation more efficient. Unlike classic MCTS methods in domains like Go or multi-agent systems that require step-wise search due to external feedback, reasoning tasks with MLLMs allow for generating an entire reasoning chain at once, reducing redundant computation and accelerating the search process.
> Additionally, our experiments show that CoMCTS significantly reduces the average search iterations compared to LE-MCTS.
> ||Le-MCTS|CoMCTS|
> |-|-|-|
> |Average Search Iteration|32.8|12.7|
> 4) Reflection design. We introduce a new method that leverages error sibling nodes to construct reflection data. To our knowledge, this paper is the first to explore the reflective capabilities of MLLMs through tree search and collective knowledge.
>
> ---
> **W2: Clarification of CoMCTS results**
>
> > The performance improvement from MCTS alone appears marginal. For instance, in Table 2, GPT-4o direct achieves 58.2, while GPT-4o + CoMCTS reaches only 63.8—a modest gain given the increased computational overhead.
>
> Thank you!
> We believe there may be some misunderstanding regarding the value of GPT-4o + CoMCTS in Tab 2.
> The improvements brought by CoMCTS primarily result from **collaborative search with multiple models**, as shown in the last row of Table 2 (58.2% -> 80.3%) and the search success rates in Table 4 .
> In Table 2, the “GPT-4o + CoMCTS” setting uses only a single model, but the MCTS configuration is identical to that of CoMCTS, such as the reasoning and error localization prompts. So we named it GPT-4o + CoMCTS. We will revise it to provide more clear clarifications and will change “GPT4o + CoMCTS” to “GPT-4o + MCTS”.
>
> > The more significant improvement seems to stem from ensembling the Qwen2-VL-72B model rather than MCTS itself.
>
> We conducted experiments comparing the search success rate using MCTS with a single GPT-4o, a single Qwen2-VL-72B, and CoMCTS with GPT-4o + Qwen2-VL-72B.
> The results below show that the search performance of individual models is limited (i.e., 63.8 for GPT-4o, 56.1 for Qwen2-VL-72B), and the main performance gain comes from collective intelligence (i.e., 74.6 for GPT-4o + Qwen2-VL-72B) rather than from a single powerful model.
> Besides, we have provided the direct prediction results using only Qwen2-VL-72B in Table 9 of Appendix D.1. These results show that the reasoning capability of Qwen2-VL-72B alone is limited (48.3), further demonstrating that the benefits of CoMCTS primarily stem from collective intelligence rather than from a single strong MLLM.
>
> ||MCTS (GPT-4o)|MCTS (Qwen2-VL-72B)|CoMCTS (GPT-4o + Qwen2-VL-72B)|
> |-|-|-|-|
> |S.S.R|63.8|56.1|74.6|
> ---
> **W3: Effectiveness of collective learning group and computation overhead**
>
> Thank you!
> > The effectiveness of CoMCTS hinges on the diversity and quality of the models in the collective learning group.
>
> We respectfully clarify that a single MLLM for reasoning is ineffective, which motivates us to leverage collective intelligence for collaborative reasoning.
> Employing stronger models for collaborative reasoning and error localization can further improve the results of CoMCTS (see our response to Reviewer H5ne’s Q2 for details), highlighting the long-term potential of collective intelligence and collaborative learning for future advances.
>
> > Additionally, the reliance on multiple models for collective learning and iterative tree search increases the computational overhead.
>
> Regarding computational overhead, inference with smaller models such as Qwen2-VL-7B is relatively fast (1s per inference), and GPT-4o API calls are also highly efficient. Although larger model Qwen2-VL-72B is more resource-intensive, utilizing vLLM for inference can significantly accelerate the process (2-3s per inference).
> While utilizing multiple models for search inevitably increases computational overhead, we believe that a more important benefit is the improvement in the quality of the generated reasoning data.
> When training the Mulberry model with data searched via CoMCTS, using the same base MLLMs and a comparable data scale, the results outperform both LLaVA-CoT and LLaVA-Reasoner, as shown in Tab 1, Tab 6, Tab 16, and in our responses to reviewer GUCu's W1 and reviewer H5ne’s Q1.
>
> ---
> **Q1: Innovations of CoMCTS and its Relation to Other MCTS Methods**
>
> Thank you!
> We would like to kindly clarify that we discuss the relationship of our paper to other MCTS methods in Appendix E.1 and to other reasoning MLLMs in Appendix E.2. Additionally, we will include the discussions from W1 in the revised version of our paper.
>
> Below, we summarize the key contributions of CoMCTS and highlight the differences compared to previous MCTS methods:
> 1) we introduce collective learning into MCTS and propose CoMCTS, which leverages collective knowledge to collaboratively conjecture, search and identify effective and reflective reasoning paths for MLLMs, significantly improving search effectiveness and efficiency. To our knowledge, this is the first work that explores collective learning with MCTS for MLLMs.
> 2) Its joint expansion mechanism allows CoMCTS to combine reasoning trajectories from multiple MLLMs through iterative search, constructing a unified reasoning tree with diverse and complementary nodes. This leverages the strengths of different models while avoiding traps in low-quality, homogeneous nodes within a single MLLM's reasoning space.
> 3) In each search iteration, CoMCTS generates a complete reasoning trajectory and selects the last correct step as the new starting node, skipping intermediate steps to significantly reduce search time without sacrificing effectiveness. Furthermore, collective knowledge enhances error localization by enabling mutual validation among models.
> 4) Using CoMCTS, we search for effective, reflective reasoning paths for multimodal inputs and construct Mulberry-260k, a dataset containing rich, explicit reasoning trees for each question.
> 5) To our knowledge, Mulberry is the first work to explore the reflective capabilities of MLLMs through tree search and collective knowledge.
> ---
> **Q2: Measuring CoMCTS search phase overhead**
>
> Thank you for your question.
> Below, we provide detailed estimates of the total token usage, GPU hours, and API costs incurred during data generation in the CoMCTS search phase.
>
> ||total token cost (M)|GPU Hours (H)|API price ($)|
> |-|-|-|-|
> |Overhead|965|6528|3200|
> ---
> **Q3: Statistics on the contribution of each model in CoMCTS**
>
> Thank you for your valuable suggestions.
> We agree that providing statistics on the contribution of each model to CoMCTS search is important and informative.
> > the similarity in sampling between different models
>
> To compute similarity, we first encode the reasoning paths generated by different models into embeddings using the all-MiniLM-L6-v2 model. We then calculate the similarity between these embeddings. The results are shown in the table below.
>
> ||GPT-4o × LLaMA3.2-V-11B|Qwen2-VL-72B × Qwen2-VL-7B|Qwen2-VL-72B × GPT-4o|Qwen2-VL-72B × LLaMA3.2-V-11B|
> |-|-|-|-|-|
> |similarity|0.8608|0.8914|0.8283|0.8014|
> > or the proportion of each model in generating correct paths
>
> In addition, we also report the contribution of each model to the generated correct paths, as shown below.
> |Models|GPT-4o|Qwen2-VL-72B|Qwen2-VL-7B|LLaMA3.2-V-11B|
> |-|--|-|-|--|
> |Proportion in generating correct paths|47|32|13|8|
>
> ---
> **Q4: CoMCTS results without smaller models**
>
> Thank you!
> > Is it possible to achieve good performance without introducing smaller models, for example, by only using GPT-4o and Qwen2-VL-72B?
>
> We agree that only using larger foundation models can achieve good performance.
> However, incorporating smaller models provides additional benefits with minimal computational overhead.
> As shown below, adding two small models leads to a 5.6% improvement in search success rates.
>
> > For Qwen2-VL-72B and Qwen2-VL-7B, their behaviors might be similar due to shared training data—does integrating both models still provide an advantage?
>
> |Models|CoMCTS (GPT-4o + Qwen2-VL-72B)|CoMCTS (GPT-4o + Qwen2-VL-72B + Qwen2-VL-7B + LLaMA3.2-V-11B)|
> |-|-|-|
> |S.S.R|74.6|80.2|
>
> The similarity analysis from Q3 also shows that while Qwen2-VL-7B and Qwen2-VL-72B generate relatively similar reasoning responses, they are not identical. They can still collaboratively contribute diverse ideas.
>
> ----
> Thank you again for your insightful and professional comment, which made our work more complete and solid! If there are any further questions, please let us know.
> If you feel all questions have been addressed, we would be grateful if you could kindly consider re-rating our work. Thanks!

---

> ### Author Response · Authors · 2025-08-05
> **Many thanks for your professional and valuable review**
>
> Thank you once again for your thoughtful review, which has greatly improved the quality and clarity of our paper.
>
> In the revised version, we will highlight the innovations of CoMCTS and the Mulberry models, and we will include a discussion of the differences between LE-MCTS and CoMCTS.
> Furthermore, we will incorporate the results and discussions from the rebuttal period into the revised paper.
> If you have any further questions or suggestions, please feel free to reach out. We will remain actively engaged in the discussion until the end of the rebuttal period.

---

### Official Review · Reviewer_GUCu · 2025-07-01

**Clarity:** 4
**Significance:** 3
**Originality:** 3
**Rating:** 5
**Confidence:** 3

**Summary:**

This study introduces Collective Monte Carlo Tree Search (CoMCTS) to equip Multimodal Large Language Models (MLLMs) with step-by-step reasoning and reflective capabilities. Unlike traditional single-node expansion, CoMCTS performs collective expansion, where each MLLM in the group simultaneously generates potential reasoning paths from the current leaf node. This collaborative expansion mechanism aims to (1) generate diverse and complementary candidate reasoning nodes, (2) prevent the search process from getting stuck in low-quality reasoning spaces of individual models, and (3) significantly improve search efficiency by exploring multiple paths in parallel. Furthermore, this study generated a high-quality tree-structured multimodal reasoning dataset (Mulberry-260k), demonstrating that training on this dataset enhances the reasoning abilities of the Mulberry model and improves performance on complex multimodal benchmarks.

**Questions:**

See the weaknesses.

**Ethical Concerns:**

["NO or VERY MINOR ethics concerns only"]

**Final Justification:**

The author response gave empirical evidence that partially addressed my concerns about the limited evaluation of MLLMs and the usefulness of collective intelligence from different MLLMs. I think this paper provides a worthy contribution to the conference.

**Limitations:**

I could not find any dedicated section of paragraph explaining the potential limitations of this work. I'd recommend adding the explanation.

**Quality:**

3

**Strengths And Weaknesses:**

Strength

+ CoMCTS enables multiple models to collaboratively expand reasoning paths and collectively identify and prune erroneous steps, thereby significantly enhancing the effectiveness and efficiency of MLLM search.
+ Training MLLMs on data generated through CoMCTS, data that includes explicit step-by-step reasoning and reflection paths, leads to improved reasoning abilities and allows the models to self-correct during complex tasks.
+ High-quality tree-structured multimodal reasoning datasets like Mulberry-260k are valuable for advancing MLLM intelligence beyond direct prediction, toward more robust, explainable, and human-like reasoning processes.

Weakness

+ Although CoMCTS itself is not specific to MLLMs, the evaluation is conducted solely on MLLMs. From the outset of Section 1, the research assumes an MLLM-based setup, but it would be necessary either to include evaluations on tasks like mathematics or to provide a justification for not addressing such tasks.
+ Table 2 demonstrates the effect of using collective intelligence from different MLLMs. However, the most significant performance gain is observed when Qwen2-VL-72B is added. Therefore, it would be interesting to see the experimental results of applying CoMCTS using only Qwen2-VL-72B. In other words, it is important to distinguish whether the improvement stems from collective intelligence or from the capabilities of the powerful MLLM, Qwen2-VL-72B.

---

> ### Author Rebuttal · Authors · 2025-07-31
>
> Dear Reviewer GUCu,
>
> Many thanks for your professional review!
> We are pleased that you have recognized our key contributions: 1) the effectiveness and efficiency of CoMCTS; 2) reasoning and self-correct capabilities of Mulberry; 3) high-quality tree-structured multimodal reasoning datasets Mulberry-260k.
> Below, we tried our best to respond to your concerns and suggestions one by one.
>
> ---
>
> **W1: Additional evaluation of CoMCTS on pure-text LLM \& More results on mathematics benchmark.**
>
> Thank you for your valuable feedback! We are going to respond to this concern in two parts:
>
> > Although CoMCTS itself is not specific to MLLMs, the evaluation is conducted solely on MLLMs.
>
> **Additional evaluation of CoMCTS on pure-text LLM.**
> As stated in Lines 29-30 and 36-37 and throughout the paper, our work is motivated by the limited reasoning capabilities of existing MLLMs, which prompted us to design CoMCTS for collaboratively searching high-quality reasoning data for MLLMs.
> Accordingly, this paper focuses on multi-modal reasoning data search and MLLM training, and our evaluations have thus been conducted on MLLMs.
> However, to address your insightful question, we conducted an experiment using 100 cases sampled from DAPO-Math, showing that collective intelligence also improves search success rates in pure text reasoning tasks using LLMs, as presented in the table below.
>
> |       | MCTS (4o) | MCTS (Qwen2.5-72B) | CoMCTS (4o+Qwen2.5-72B) |
> | ----- | --------- | ------------------ | ----------------------- |
> | S.S.R | 25\%      | 18\%               | 39\%                    |
>
>
> > From the outset of Section 1, the research assumes an MLLM-based setup, but it would be necessary either to include evaluations on tasks like mathematics or to provide a justification for not addressing such tasks.
>
> **More results on mathematics benchmark.**
> We agree that evaluating mathematics tasks is both necessary and important for assessing the reasoning abilities of MLLMs.
> To this end, we would like to kindly clarify that we have reported the performance of Mulberry models on mathematics benchmarks such as MathVista, DynaMath and MM-Math in Table 1.
> In response to your suggestion, we have further expanded our evaluation to include additional mathematics benchmarks, including MathVision, MathVerse and WeMath, as shown in the table below.
> The results show that Mulberry models consistently achieve substantial improvements over base models and reasoning MLLMs across all evaluated mathematics benchmarks.
> We believe these results further demonstrate the effectiveness of our approach.
> Thank you!
> We will include these results in the revised version.
>
> | Models                        | MathVista | DynaMath | MM-Math | MathVision | MathVerse | WeMath | AVG  |
> |-------------------------------|-----------|----------|---------|-------------|-----------|--------|------|
> | LLaMA-3.2-V-11B (Base Model)  | 48.6      | 34.3     | 4.1     | 19.3        | 26.9      | 34.6   | 27.9 |
> | LLaVA-CoT-11B                 | 54.8      | 36.8     | 10.3    | 20.2        | 33.4      | 39.3   | 32.5 |
> | Mulberry-11B                  | 61.1      | 37.2     | 18.7    | 26.0        | 36.1      | 44.3   | 37.2 (**+9.3%**) |
>
> ----
>
> **W2: Effect of collective intelligence vs. single powerful MLLM (Qwen2-VL-72B).**
>
> Thank you for raising this concern.
> We conducted experiments comparing the search success rate on 100 samples using MCTS with a single GPT-4o, a single Qwen2-VL-72B, and CoMCTS with GPT-4o + Qwen2-VL-72B.
> The results below show that the search performance of individual models is limited (i.e., 63.8 for GPT-4o, 56.1 for Qwen2-VL-72B), and the main performance gain comes from collective intelligence (i.e., 74.6 for GPT-4o + Qwen2-VL-72B) rather than from a single powerful model.
> Additionally, we have provided the direct prediction results using only Qwen2-VL-72B in Table 9 of Appendix D.1. These results also show that the reasoning capability of Qwen2-VL-72B alone is limited (48.3), further demonstrating that the benefits of CoMCTS primarily stem from collective intelligence rather than from a single strong MLLM.
>
> | Models | MCTS (GPT-4o) | MCTS (Qwen2-VL-72B) | CoMCTS (GPT-4o + Qwen2-VL-72B) |
> |--------|---------------|---------------------|--------------------------------|
> | S.S.R  | 63.8          | 56.1                | 74.6                           |
>
>
> ---
>
> Lastly, thank you very much for your constructive feedback and suggestions.
> We will incorporate these points into the revised paper.
> If you have any further questions, please feel free to let us know. We will be available throughout the rebuttal period.

---

> > ### Comment · Reviewer_GUCu · 2025-08-06
> > **Re: Rebuttal by Authors**
> >
> > Thank you for the detailed response. My concerns are partially addressed, and I will raise the rating.

---

> > > ### Author Response · Authors · 2025-08-07
> > > **Many thanks for your professional and helpful review**
> > >
> > > We sincerely thank you for your constructive review. Your questions and suggestions have greatly improved the clarity and quality of our paper. We will incorporate the discussions and additional experiments from the rebuttal period into the revised version. Thank you again.

---

### Official Review · Reviewer_H5ne · 2025-07-03

**Clarity:** 3
**Significance:** 3
**Originality:** 3
**Rating:** 5
**Confidence:** 4

**Summary:**

This paper develops Mulberry, a Multimodal reasoning model based on Collective Monte Carlo Tree Search. Specifically, the authors introduce collective learning into “tree search” for reasoning-path searching and learning, construct Mulberry-260k dataset, and then finetune a series of multimodal models, aiming to empower the models with reasoning ability. Experiments demonstrate the effectiveness of the models.

**Questions:**

1. Table 1 has too many missing values.

2. For Table 2's comparison, what results might emerge if all four models were strong foundation models (e.g., GPT-4o)?

3. Would the authors consider open-sourcing the training data and models?

4. Some writing problems: L68 has an extra space. It is recommended to use \citet{} on L123-125.

**Ethical Concerns:**

["NO or VERY MINOR ethics concerns only"]

**Final Justification:**

The authors provided a clear and detailed rebuttal addressing my main concerns. While I initially felt the contribution was somewhat incremental, the rebuttal provided sufficient justification of novelty, especially around the collective learning mechanism in CoMCTS and its integration with MLLMs. I have accordingly raised my rating to 5 (Weak Accept).

**Quality:**

3

**Strengths And Weaknesses:**

Strengths:
1. The paper is well organized and easy to follow.
2. This work introduces the Mulberry-260K dataset and Mulberry models, which will facilitate future research in multimodal reasoning and related areas.
3. Research findings indicate that multimodal models, even with small sizes (2B parameters), can benefit from the reasoning and reflective abilities.



Weakness:
While this work provides a solid incremental contribution, the novelty and insights appear somewhat limited.

---

> ### Author Rebuttal · Authors · 2025-07-31
>
> Dear Reviewer H5ne,
>
> We appreciate the reviewer’s valuable comments and feedback.
> Thank you for recognizing the clear organization of this paper and for highlighting the contributions of the Mulberry-260K dataset and Mulberry models to future research in multimodal reasoning.
> We are also grateful for your recognition that even small multimodal models can benefit from enhanced reasoning and reflective abilities.
> We have carefully responded to each of your concerns and suggestions below, and will revise the manuscript accordingly.
>
> ----
>
> **W1: The novelty and insights of CoMCTS and Mulberry Models.**
>
> Thank you for your question!
> We believe our work, including CoMCTS and Mulberry Models, offers unique innovations.
> Below, we first summarize the key contributions of our paper, and then highlight the differences between CoMCTS and other MCTS-based methods, as well as between Mulberry and other reasoning MLLMs.
>
> **Contributions**:
> **First**, we introduce collective learning into MCTS and propose CoMCTS, which leverages collective knowledge to collaboratively conjecture, search and identify effective and reflective reasoning paths for MLLMs, significantly improving search effectiveness and efficiency.
> To our knowledge, this is the first work that explores collective learning with MCTS for MLLMs.
> **Second**, we construct Mulberry-260K, a valuable resource for step-by-step reasoning and reflection research in MLLMs.
> **Third**, we develop Mulberry, a series of MLLMs with strong step-by-step reasoning and reflection abilities.
> **Fourth**, extensive experiments validate the superiority of our methods across various benchmarks.
>
> **Relation to Other MCTS Methods**:
> We summarize the contributions of CoMCTS and highlight the differences compared to previous MCTS methods (e.g., Traditional MCTS, ReST-MCTS):
> (a) CoMCTS introduces collective learning into tree search for MLLMs, enabling more effective and efficient reasoning path searching and learning.
> (b) Its joint expansion mechanism allows CoMCTS to combine reasoning trajectories from multiple MLLMs through iterative search, constructing a unified reasoning tree with diverse and complementary nodes. This leverages the strengths of different models while avoiding traps in low-quality, homogeneous nodes within a single MLLM's reasoning space.
> (c) In each search iteration, CoMCTS generates a complete reasoning trajectory and selects the last correct step as the new starting node, skipping intermediate steps to significantly reduce search time without sacrificing effectiveness. Furthermore, collective knowledge enhances error localization by enabling mutual validation among models.
>
> **Relation to Other Reasoning MLLMs**:
> We summarize the contributions of Mulberry models and highlight the differences compared to previous reasoning MLLMs:
> (a) Using CoMCTS, we search for effective, reflective reasoning paths for multimodal inputs and construct Mulberry-260k, a dataset containing rich, explicit reasoning trees for each question, which is then used to train the Mulberry models.
> (b) Unlike pre-defined stage-based methods (e.g., LLaVA-CoT, LLaVA-Reasoner), CoMCTS searches step-level reasoning data of varying lengths, enabling Mulberry to perform flexible step-by-step reasoning.
> (c) To the best of our knowledge, Mulberry is the first work to explore the reflective capabilities of MLLMs through tree search and collective knowledge.
>
> We hope these clarifications help to highlight our contributions and the novelty of our work.
>
> ----
>
> **Q1: Supplementing the values in Table 1.**
>
> Thank you for your suggestions.
> Some values are missing because the original papers did not report results on certain benchmarks, which resulted in incomplete values in our Table 1.
> Many thanks for your feedback. We have now conducted additional evaluations on these models ourselves to provide a more comprehensive comparison.
> During the rebuttal period, we present a part of results in the table below, which shows that the Mulberry models outperform other reasoning MLLMs across the overall benchmarks using same base MLLMs.
> The remaining missing values will be included in the revised paper.
>
> We believe offering these missing values would improve the completeness of our comparisons. Thanks!
>
> | Models                | MathVista | MMStar | MMMU  | ChartQA | DynaMath | HallBench | MM-Math | MME_sum | AVG   |
> |-----------------------|-----------|--------|-------|---------|----------|------------|---------|---------|-------|
> | LLaVA-NeXT-8B         | 37.5      | 42.1   | 41.7  | 69.5    | 22.7     | 33.4       | 0.6     | 1957    | 39.7  |
> | LLaVA-Reasoner-8B     | 50.6      | 54.0   | 40.0  | 83.0    | 30.1     | 32.7       | 6.2     | 2011    | 46.1  |
> | Mulberry-8B           | 56.3      | 54.5   | 43.0  | 79.5    | 34.1     | 47.5       | 18.9    | 2021    | 50.7 (**+11**) |
>
> | Models                | MathVista | MMStar | MMMU  | ChartQA | DynaMath | HallBench | MM-Math | MME_sum | AVG   |
> |-----------------------|-----------|--------|-------|---------|----------|------------|---------|---------|-------|
> | Llama-3.2-11B-V-Ins.  | 48.6      | 49.8   | 41.7  | 83.4    | 34.3     | 40.3       | 4.1     | 1787    | 45.8  |
> | LLaVA-CoT-11B         | 54.8      | 57.6   | 44.9  | 82.1    | 36.8     | 47.8       | 10.3    | 2077    | 51.1  |
> | Mulberry-11B          | 61.1      | 58.5   | 45.6  | 83.5    | 37.2     | 48.9       | 18.7    | 2035    | 53.3 (**+7.5**) |
>
> -----
>
> **Q2: CoMCTS results with four strong foundation models.**
>
> Thank you for your question!
> In the table below, we present results of CoMCTS using four strong foundation MLLMs (GPT-4o, Claude-3.5 Sonnet, Qwen2-VL-72B and InternVL2-40B).
> As model strength increases, CoMCTS is able to generate more accurate reasoning steps and error localization, resulting in higher search success rates.
> These findings also highlight the long-term potential and scalability of CoMCTS as foundation models become increasingly powerful.
> We will include these discussions in the revised paper.
>
> | Models | CoMCTS (GPT-4o + Qwen2-VL-72B + LLaMA3.2-V-11B + Qwen2-VL-7B) | CoMCTS (GPT-4o + Qwen2-VL-72B + Claude-3.5 Sonnet + InternVL2-40B) |
> |--------|:--------------------------------------------------------------:|:--------------------------------------------------------------------:|
> | S.S.R  | 80.2                                                         | 88.9                                                               |
>
> ----
>
> **Q3: Open source CoMCTS code, data, and models.**
>
> Thank you for raising this point.
> We would like to clarify that we will release all resources, including the CoMCTS search code, training data, training scripts, model checkpoints, and evaluation code, to facilitate reproducibility and further research.
>
> ----
>
> **Q4: Addressing writing problems at L68 and L123–125.**
>
> Thank the reviewer for pointing out these writing issues, which help improve the clarity and quality of our paper.
> We have removed the extra space at L68 and revised the citation in L123–125 to use \cite{} as recommended in the revision.
>
> ---
>
> Lastly, thank you so much for helping us improve the paper and we appreciate your open discussions! Please let us know if you have any further questions. We are actively available until the end of this rebuttal period. Looking forward to hearing back from you!

---

> > ### Comment · Reviewer_H5ne · 2025-08-04
> >
> > Thank you for the clarifications and additional experiments. I raise my rating to 5.

---

> ### Author Response · Authors · 2025-08-05
> **Many thanks for your professional review!**
>
> Thank you again for your review, which has greatly improved the quality of our paper.
>
> In the revised version, we will highlight the novelty and insights of CoMCTS and the Mulberry models, and we will include all additional experimental results. Furthermore, we will open-source all resources to support and facilitate future research.
> Please feel free to reach out if you have any further questions or suggestions.

---

### Official Review · Reviewer_Wx49 · 2025-07-04

**Clarity:** 4
**Significance:** 3
**Originality:** 3
**Rating:** 5
**Confidence:** 5

**Summary:**

The paper introduces Collective Monte Carlo Tree Search (CoMCTS), an ensemble-style extension of MCTS that aggregates proposals from multiple MLLMs. Aside from the usage of multiple models for generating and evaluating the steps, another innovation is to use some wrong nodes in the chain to simulate self-reflection via transitioning to the right step. CoMCTS improves the search success rate and reduces the number of iterations over standard MCTS. A large scale dataset is created for finetuning MLLMs, which leads to substantial performance improvements in several MLLMS on multiple standard benchmarks.

Overall, this paper presents a significant advancement in search based multimodal reasoning by combining large model ensembles with MCTS and reflective data generation. The proposed approach is innovative, practical, and well-validated empirically, and the dataset in itself is a valuable contribution. Therefore, I recommend acceptance.

**Questions:**

I do not have any major questions, please address the concerns raised in the 'Weaknesses' section.

**Ethical Concerns:**

["NO or VERY MINOR ethics concerns only"]

**Final Justification:**

I thank the authors for their response.

After rebuttal: W1. The authors show that although CoMCTS uses more tokens/sample compared to LLaVA-CoT, it results in a considerably higher search success rate, this addresses my question about the computational requirement and tradeoff between token usage and efficacy. W2. minor suggestion, addresses satisfactorily.

Overall, after reading the other reviews and rebuttals, my assessment of the paper has not changed considerably (if anything, the authors' rebuttals to other reviewers' questions strengthen the paper). Therefore, I keep my score to indicate an Accept.

**Limitations:**

Addressed adequately in Appendix K.

**Quality:**

4

**Strengths And Weaknesses:**

Strengths:

 1) A new large-scale multimodal dataset is constructed, annotated with structured reasoning trees, error paths, and reflection nodes. This provides rich training signals for developing compositional reasoning abilities in MLLMs.

2) Mulberry demonstrates strong results across multiple multimodal reasoning benchmarks, outperforming open-source models and rivaling some closed-source systems.

3) Ablations clearly demonstrate the utility of each CoMCTS component. The use of erroneous nodes encourages models to self-correct and engage in more human-like, iterative thinking, and combining steps from multiple models is more effective than using each of those individual models.


Weaknesses:

1) Running multiple MLLMs in parallel and pruning through simulations remains expensive. A comparison of necessary token usage for creating LLAVA CoT-100k and this dataset would be useful in understanding this aspect thoroughly.

2) The organization of the experimental results could be better. For example, the contents in Appendix C.1 and C.3 are very interesting and it would possibly be better if they are included in the main paper. Otherwise Table 1 gives the impression that Mulberry is only compared to the base models which is not true. Highlighting its effectiveness in comparison to other ways of finetuning the MLLMs would strengthen the paper.

---

> ### Author Rebuttal · Authors · 2025-07-31
>
> Dear Reviewer Wx49,
>
> We appreciate your valuable comments and suggestions.
> We are encouraged by your recognition of our contributions in constructing a large-scale multimodal reasoning dataset, the strong performance of the Mulberry models, and the clear utility of each CoMCTS component shown in our ablations.
> We have carefully considered all your suggestions and will revise the manuscript to include a more detailed discussion on token usage and improve the overall organization as suggested.
>
> ---
>
> **W1: Comparison of necessary token usage.**
>
> Thank you for your suggestion.
> We agree that including a comparison of token usage will help provide a more thorough understanding of this area.
> Therefore, we provide a comparison between CoMCTS and other CoT data construction methods regarding search token usage per question, search success rate, and token usage per sample for successfully finding the correct answer, as shown in the table below.
> Although CoMCTS consumes more tokens during the search phase, it achieves higher search success rates and data quality, leading to superior performance of the Mulberry models.
>
>
> | Models                     | Token usage per sample | Search success rate (%) | Token usage per sample for successfully finding the correct answer |
> |----------------------------|------------------------|--------------------------|----------------------------------------------------------------------|
> | MCTS                       | 10796                  | 63.8                     | 16922                                                                |
> | LLaVA-CoT                  | 212.4                  | 57.1                     | 404                                                                  |
> | BoN (N=10) + LLaVA-CoT     | 2184                   | 61.3                     | 3465                                                                 |
> | CoMCTS                     | 2977                   | 80.2                     | 3713                                                                 |
>
> | Models | Dataset construction method | Avg accuracy |
> |---------|----------------------------|-------------------|
> | LLaVA-Reasoner-8B  | Direct prediction  | 46.1%            |
> | Mulberry-8B  | CoMCTS | 50.7% (**+4.6**)                        |
> | LLaVA-CoT-11B  | Direct prediction  | 51.1%                        |
> | Mulberry-11B  | CoMCTS | 53.3%  (**+2.2**)                    |
>
>
> Additionally, we manually evaluated the correctness of reasoning paths in 100 cases generated by CoMCTS and the direct prediction method (e.g., LLaVA-CoT). The results show that by leveraging collective error localization, CoMCTS significantly improves the accuracy of intermediate reasoning steps.
>
> | Methods | Reasoning Path Error Rate |
> |---------|----------------------------|
> | Direct prediction   | 51%                        |
> | CoMCTS  | 18%                        |
>
> ----
>
> **W2: Better organization of the experimental results.**
>
> Thank you very much for your valuable suggestion.
> We agree that incorporating the contents from Appendix C.1 and C.3 into the main paper could improve the clarity and comprehensiveness of our experimental comparisons.
> In the revised version, we will move the results and analyses from Appendix C.1 and C.3 into the main text, and we will update Table 1 to clearly indicate that Mulberry is compared with both base models and other reasoning MLLMs using the same backbone.
> We believe these changes help to better highlight the effectiveness of our approach and address your concern.
>
> ----
>
> Many thanks for your professional, detailed, and valuable reviews! Please let us know if you have any other questions. We will actively join the discussion until the end of the rebuttal period.

---

> > ### Comment · Reviewer_Wx49 · 2025-08-04
> > **maintaining my positive assessment**
> >
> > I thank the authors for addressing my concerns. I am maintaining my score.

---

> ### Author Response · Authors · 2025-08-05
> **Thank you for your valuable and insightful review.**
>
> We're glad to hear that your concerns have been adequately addressed. We appreciate your professional and constructive feedback which made our work more solid and clear.
>
> We'll revise the paper based on our discussions to better present Mulberry. If you have any questions or suggestions, please feel free to comment here. Thank you.

---

### Decision · Program_Chairs · 2025-09-17

**Decision:**

Accept (spotlight)

**Comment:**

This paper introduces Collective Monte Carlo Tree Search (CoMCTS), an ensemble-style extension of MCTS that equips MLLMs with step-by-step reasoning and reflective capabilities. By allowing multiple models to collaboratively expand and refine reasoning paths, and by incorporating erroneous nodes as reflective signals, CoMCTS effectively improves both the success rate and efficiency of the search process. The authors further contribute Mulberry-260k, a large-scale dataset with structured reasoning trees and reflection nodes, which significantly enhances the reasoning ability of models and leads to strong empirical results on multimodal benchmarks. Overall, I would recommend this paper for spotlight, as it not only presents a strong empirical contribution but also opens up new directions for exploring reasoning methods in MLLMs. This work highlights the potential of ensemble-style search and reflective data generation, offering an alternative paradigm to the currently dominant distillation-based approaches, and it encourages the community to experiment with more diverse ways of constructing multimodal reasoning data.